# Spontaneously established reverse electric field to enhance the performance of triboelectric nanogenerators via improving Coulombic efficiency

Yikui Gao[1,2], Lixia He[1,2], Di Liu [1,2] ✉, Jiayue Zhang [3], Linglin Zhou [1,2], Zhong Lin Wang [1,2,4,5] ✉ & Jie Wang [1,2] ✉

Mechanical energy harvesting using triboelectric nanogenerators is a highly desirable and sustainable method for the reliable power supply of widely distributed electronics in the new era; however, its practical viability is seriously challenged by the limited performance because of the inevitable side-discharge and low Coulombic-efficiency issues arising from electrostatic breakdown. Here, we report an important progress on these fundamental problems that the spontaneously established reverse electric field between the electrode and triboelectric layer can restrict the side-discharge problem in triboelectric nanogenerators. The demonstration employed by direct-current triboelectric nanogenerators leads to a high Coulombic efficiency (increased from 28.2% to 94.8%) and substantial enhancement of output power. More importantly, we demonstrate this strategy is universal for other mode triboelectric nanogenerators, and a record-high average power density of 6.15 W m$^{-2}$ Hz$^{-1}$ is realized. Furthermore, Coulombic efficiency is verified as a new figure-of-merit to quantitatively evaluate the practical performance of triboelectric nanogenerators.

Sustainable and reliable power sources that are not detrimental to the environment are essential and in great demand in the new era of distributed sensor networks, artificial intelligence (AI), and internet of things (IoTs). The potential power sources are expected to enable continuous, long-term, and self-powering the widely distributed electronics and to facilitate future remote monitoring for intelligent sensing[1,2]. Due to clean, renewable, abundant, and ubiquitous advantages, mechanical energy is considered to be a particularly attractive source[3,4]. An advanced energy harvesting device that generates electrical power from mechanical motion could be an important step towards next-generation self-powered systems[5].

Among various mechanical energy harvesting technologies, triboelectric nanogenerator (TENG), has attracted much more attentions due to their merits of low cost, simple structure, wide choice of materials, easy adaptive and even high efficiency at low frequency, showing promising applications in energy and sensor fields[6–8]. To realize long-term operation and enlarge the application scenario, improving the charge density of TENG is the key direction because of the square relationship between charge density and power density[9–12]. However, it has been recently revealed that electrostatic breakdown leads to a substantial charge decline on the dielectric surface and then the square loss of power density of TENGs[13,14], and that's why the

[1]Beijing Institute of Nanoenergy and Nanosystems, Chinese Academy of Sciences, Beijing 101400, P.R. China. [2]School of Nanoscience and Technology, University of Chinese Academy of Sciences, Beijing 100049, P.R. China. [3]Department of Mechanical Engineering, Tsinghua University, Beijing 100084, P. R. China. [4]Georgia Institute of Technology, Atlanta 30332, USA. [5]Yonsei Frontier Lab, Yonsei University, Seoul 03722, Republic of Korea.
✉e-mail: liudi@binn.cas.cn; zhong.wang@mse.gatech.edu; wangjie@binn.cas.cn

output charge density of TENG can be greatly improved by restricting the electrostatic breakdown in many conventional approaches, including strict environmental control[15–19] and thinner triboelectric layer design[20–22]. Besides, to evaluate and compare the performance of TENGs, the standards for quantifying the performance of TENGs were established by the cycle for maximized energy output[23–25] (denoted as CMEO). However, this context makes it difficult to understand that the actually measured average power density is largely smaller than the theoretical result calculated based on CMEO and short-circuit output charge. The core issue is that the charge loss from electrostatic breakdown (especially side-discharge) at large load or for energy storage is neglected, making the high charge density at short-circuit conditions meaningless whether for practical power management or energy storage (Supplementary Figs. 1 and 2 and Supplementary Note 1), i.e., with the electrostatic breakdown, the charge utilization efficiency under the condition of external load is dramatically lower than that without electrostatic breakdown, that's why the practical output energy is largely smaller than the theoretical result calculated based on CMEO (Supplementary Fig. 3 and Supplementary Note 2). Overall, these have presented enormous challenges: how to avoid the serious side-discharge problem to increase the output energy density while maintaining high charge utilization efficiency of TENG, and correctly evaluate its practical performance.

Here, we propose a simple but effective strategy to address the serious and troublesome side-discharge problem by introducing the SEREF on the insulator between the electrode and triboelectric layer in TENGs. Our theoretical and experimental results indicate that the additive of an insulator alongside the edge of metal electrode limits the electric field intensity around the electrode edge below the threshold of electrostatic breakdown, therefore the side-discharge problem is avoided owing to the SEREF on the insulator during the sliding of TENGs. Meanwhile, we introduce a new figure-of-merit, named Coulombic efficiency (the charge utilization efficiency of TENG under a fixed load or voltage) to correctly evaluate TENG's performance considering the issue of electrostatic breakdown, which also can be used for guiding the optimization of TENG's performance. With the demonstration of direct-current TENG (DC-TENG) devices, our strategy not only improves the short-circuit charge ($Q_{SC}$) and open-circuit voltage ($V_{OC}$), but also solves the issue of low charge utilization efficiency, and then a substantial enhancement of average power density of 2.3 W m$^{-2}$ Hz$^{-1}$ (increased by 54 times) is achieved owing to the improvement of Coulombic efficiency from 28.2% to 94.8%. More importantly, our strategy and the proposed Coulombic efficiency are feasible and universal for conventional alternate-current TENG (AC-TENG), and a record-breaking average power density of 6.15 W m$^{-2}$ Hz$^{-1}$ is realized (increased by 22 times). The strategy and Coulombic efficiency provided here set the foundation for realizing a high-performing TENG device capable of producing electricity approaching the Coulombic efficiency limit and the further industrialization of TENG technology.

## Results

### Concept of the spontaneously established reverse electric field

Electrostatic breakdown is a sudden current flow of electricity between two charged materials, which is commonly considered a negative effect in electronic industry because of the electrostatic discharge failure problem in electronic components and integrated circuits[26]. Similarly, it is often a negative effect in TENGs because electrostatic breakdown causes energy loss[9]. Conventional methods to break through the restriction of air breakdown on the upper limit of surface charge density (SCD) between two triboelectric materials focus on regulating the environmental condition including high vacuum environment, high-pressure gas, and high electrical insulation gas[15,18] (such as $SF_6$ and $N_2$, etc.) (Fig. 1a). Although the critical breakdown electric field can be heightened to a higher value, electrostatic breakdown will

take place again when the electric field strength between two tribo-electric materials exceeds $E_{B2}$ with the SCD gradual increase ($E_{B1}$ and $E_{B2}$ represent the threshold of electrostatic breakdown at air or other atmospheres; the process 2 in Fig. 1a). More importantly, implementing TENG to work in a special environmental condition requires strict encapsulation, which deprives TENG's advantage of flexibility and limits its practical application.

Different from conventional methods, here we propose a strategy to suppress electrostatic breakdown by decreasing the electric field strength between electrode and triboelectric layer. The specific method is carried out by only pasting an insulator at the electrode edge (Fig. 1b <ii>), which is used to prevent charge leakage due to side-discharge flow into the electrode, and accumulate static charges on the side surface of insulator to establish a reverse electric field (Fig. 1c). For sliding triboelectrification, negative charges often transfer from the metal electrode to the triboelectric layer based on triboelectrification, and then electrostatic breakdown preferentially occurs around the electrode edge because the electric field intensity between the electrode edge and triboelectric layer (TL) is always highest in that of the TL interface (Fig. 1d <i> and Fig. 1e <i>). When an insulator is pasted on the electrode edge (Fig. 1b <ii>), the simulated results indicate that the electric field intensity around the electrode edge also exceeds the breakdown threshold (Fig. 1d <ii> and Fig. 1e <ii>). However, massive negative charges will accumulate on the surface of insulator due to electrostatic breakdown during sliding movement (Fig. 1c and Supplementary Fig. 4), and it can be maintained at a high value for a period of time without external charge supplementation (Supplementary Fig. 5). It is clearly that the established electric field by the charged insulator is opposite to that of triboelectric layer (Supplementary Fig. 6), which weakens the electric field around electrode edge and suppresses subsequent electrostatic breakdown (Fig. 1d <iii> and Fig. 1e <iii>). More importantly, the spontaneously established reverse electric field (SEREF) on insulators is not a fixed value. As the SCD on triboelectric layer increases, the SCD of insulator also increases due to more severe electrostatic breakdown, and then a stronger SEREF is established to avoid the recurrence of electrostatic breakdown, realizing the self-regulation of SEREF (the process 3 in Fig. 1a, Supplementary Fig. 7). With the SEREF on the insulator introduced to a TENG, more triboelectric charges are permitted to hold on the triboelectric layer and then to be used for building high-performance TENGs. As shown in Fig. 1f, conventional strategies (increasing the threshold of electrostatic breakdown from $E_{B1}$ to $E_{B2}$) optimized TENG's performance from two aspects: $Q_{SC}$ or $V_{OC}$. Our strategy focuses on modulating the electric field intensity in the breakdown domain by the SEREF, and comprehensively optimizes TENG's performance from three aspects: $Q_{SC}$, $V_{OC}$, and Coulombic efficiency ($\eta(V)$), therefore the output energy can be greatly enhanced. The detailed discussions are shown in the following parts.

### SEREF for regulating breakdown domains of DC-TENG

According to the differences in working principles, TENGs can be divided into two categories: alternate-current TENG[14] (AC-TENG) and direct-current TENG[27] (DC-TENG) (Supplementary Note 3). Considering the unique physical and circuit models of DC-TENG, we chose the DC-TENG to demonstrate the feasibility of our strategy firstly (Fig. 2a), which can clearly exhibit the side-discharge phenomena. Based on the triboelectrification effect between the friction electrode (FE) and triboelectric layer (TL), negative charges and positive charges are generated on the surface of TL and FE (Fig. 2a <i>). When DC-TENG moves left, a unidirectional electric field will be built between charge collection electrode (CCE) and TL to induce electrostatic breakdown, and negative charges transfer from the surface of TL to CCE driven by the Coulomb force; due to the significant potential difference between CCE and FE (Fig. 2a <ii> and Supplementary Fig. 8a), negative charges will transfer from CCE to FE, generating DC output in external circuit. If

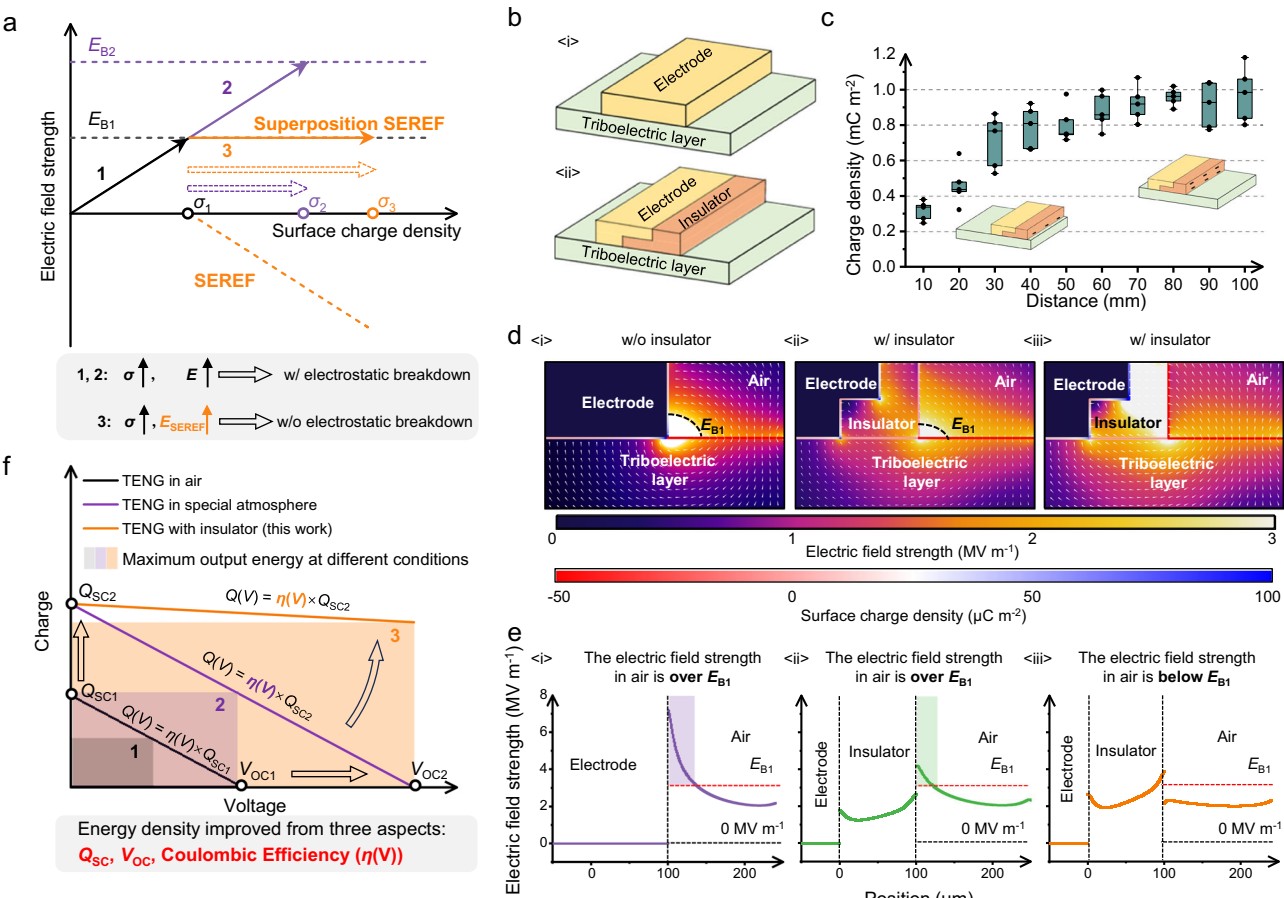

**Fig. 1 | Concept for the spontaneously established reverse electric field to restrict the side-discharge problem and its characteristics. a** 1. With the SCD increases from 0 to $\sigma_1$, the electric field strength increases from 0 to $E_{B1}$. $E_{B1}$ is the threshold of electrostatic breakdown at air. $\sigma_1$ is the corresponding maximum SCD. 2. For the special atmosphere condition (such as $SF_6$ and $N_2$, etc.), the electric field strength can increase from $E_{B1}$ to $E_{B2}$, and the corresponding maximum SCD is $\sigma_2$. 3. The SEREF on insulator can increase with the SCD, which can maintain the electric field strength of electrode edge around $E_{B1}$ and avoid the recurrence of electrostatic breakdown. **b** Schematic diagram of electrode <i> without the insulator (w/o insulator) and <ii> with the insulator (w/ insulator). **c** The SCD of insulator's side surface rapidly accumulates and reaches to 1 mC m$^{-2}$ when the moving distance is 10 cm. This box plot presents the distribution of five sets of data, with the whiskers indicating the most extreme values while excluding outliers. **d**, **e** The simulated electric field around the edge of electrode (SCD of the triboelectric layer is −50 μC m$^{-2}$.). <i> w/o insulator; the electric field at electrode edge is over $E_{B1}$ ($E_{B1}$ = 3 MV m$^{-1}$). <ii> w/ insulator; the electric field at electrode edge is also over $E_{B1}$. <iii> w/ insulator and SCD of the insulator is only −50 μC m$^{-2}$; the electric field around the edge of electrode is below $E_{B1}$. The data in **e** is taken from the electric field intensity at 5 μm above the triboelectric layer in **d**. **f** The Q-V curve represents the output charge ($Q(V)$) when the output voltage is $V$. 1. The black line is the Q-V curve of TENG in air. 2. The purple line is the Q-V curve of TENG in special atmosphere, which increases the performance of TENG from two aspects: $Q_{SC}$ and $V_{OC}$. 3. The orange line is the Q-V curve of TENG with insulator. The SEREF on the insulator can improve the performance of TENG from three aspects: $Q_{SC}$, $V_{OC}$, and $\eta(V)$. Source data are provided as a Source data file.

the DC-TENG continues to move toward the left, a continuous DC output can be obtained. When DC-TENG moves right, the electric field between CCE and TL cannot induce electrostatic breakdown because the surface charge of TL below CCE is nearly zero (Fig. 2a <iii>); thus, there is no charge transfer between CCE and FE. The detailed schematic diagram of charge transfer in DC-TENG can also be found in many works (Supplementary Fig. 8b)[26,28]. The corresponding output charge and current are shown in Fig. 2b.

According to the physical model of DC-TENG, there are three breakdown domains: the first breakdown domain (1st BD) between TL and CCE, the second breakdown domain (2nd BD) between FE and TL, and the third breakdown domain (3rd BD) between CCE and FE (Fig. 2a and Supplementary Note 4). Although our previous work indicated that suppressing the electrostatic breakdown of 2nd BD can improve the performance of DC-TENG (Supplementary Fig. 9)[29,30], there are still no detailed research on the following aspects: (1) how to suppress electrostatic breakdown; (2) the mechanism of suppressing electrostatic breakdown; (3) which parameters affect the effectiveness of suppressing electrostatic breakdown; (4) how to improve TENG's

performance by suppressing electrostatic breakdown. These questions will be detailed answered in this work.

To suppress the electrostatic breakdown of 2nd BD, here an insulator is introduced to past at the FE edge (Fig. 2c), which can utilize side-discharge around FE to accumulate charges on insulator's surface and then generates a reverse electric field to suppress electrostatic breakdown in 2nd BD. The cross section of DC-TENG is shown in Supplementary Fig. 10. The equivalent circuit (Fig. 2d, e) and corona discharge phenomena (Fig. 2f and Supplementary Fig. 11) indicate different charge flowing processes of the DC-TENG at two conditions: with insulator (w/ insulator) and without insulator (w/o insulator). For the normal DC-TENG (w/o insulator), 1st BD and 2nd BD produce corona discharge at the same time (Fig. 2f<i>), which indicates that triboelectric charges flow to TL through the 1st BD and 2nd BD, respectively (Fig. 2d<i>). The load resistance ($R_{load}$) is on the branch of the 1st BD, thus the charges flowed through 2nd BD cannot be utilized. When the insulator is pasted on the edge of FE, the corona discharge of 2nd BD disappears (Fig. 2f<ii>) and almost all triboelectric charges flow to TL through the 1st BD (Fig. 2d<ii>).

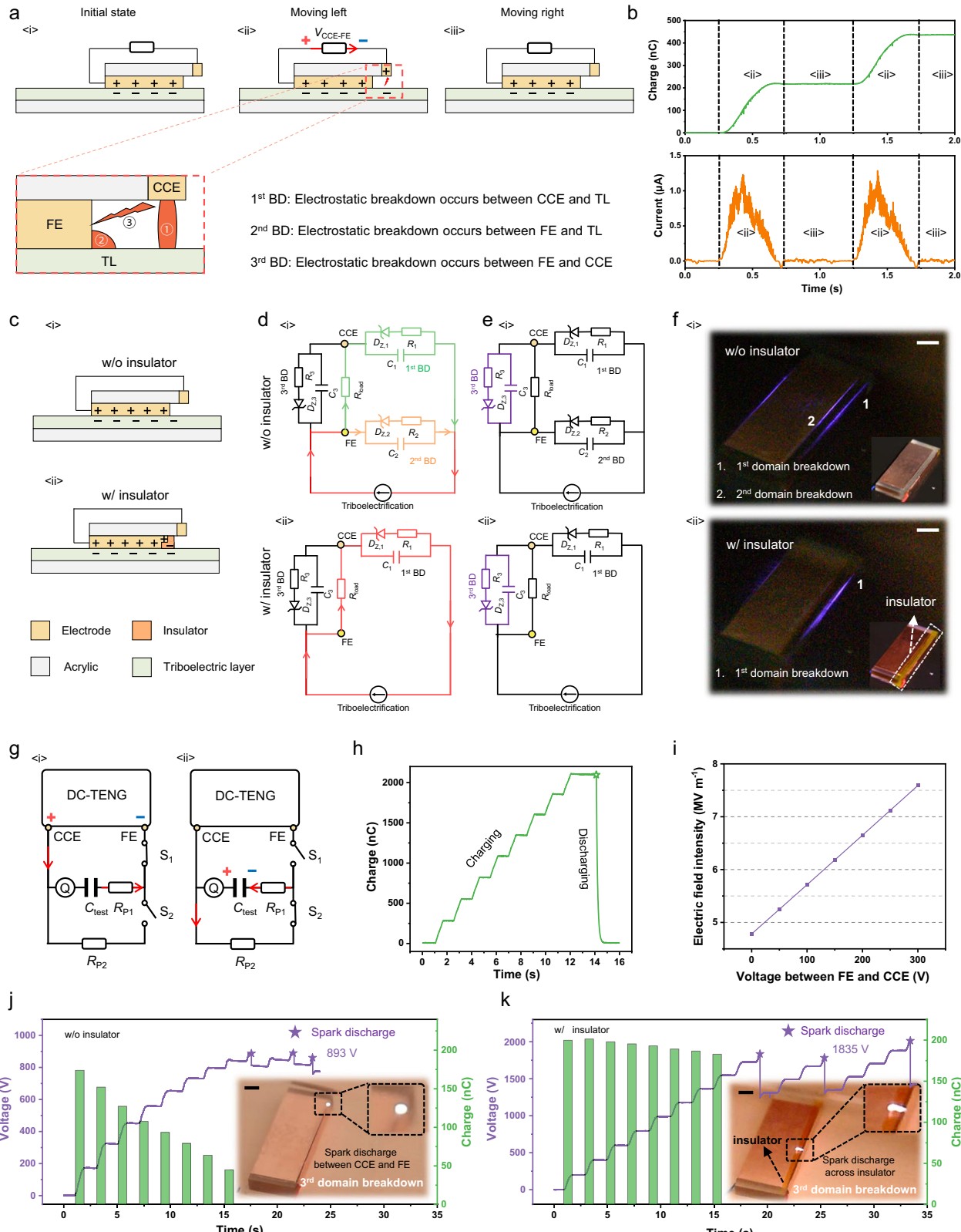

**Fig. 2 | Spontaneously established reverse electric field for regulating breakdown domains of DC-TENG. a** Schematic diagram of DC-TENG. There are three breakdown domains: 1st BD, 2nd BD and 3rd BD. **b** $Q_{SC}$ and $I_{SC}$ of DC-TENG. **c** The structure of DC-TENG <i> w/o insulator and <ii> w/ insulator. **d**, **e** The equivalent electric circuit of DC-TENG. **d** <i> DC-TENG w/o insulator: charge returned to TL through 1st BD and 2nd BD. **e** <i> DC-TENG w/ insulator: charge return to TL only through 1st BD. **d** <ii> and **e** <ii> When $V_{CCE-FE}$ increases to the threshold of 3rd BD, charge released from 3rd BD. **f** Corona discharge of 1st BD and 2nd BD in DC-TENG

<i> w/o insulator and <ii> w/ insulator (Scale bar: 5 mm). **g** The testing circuit for measuring the output charge of DC-TENG under different output voltage ($V_{CCE-FE}$). $R_{P1}$ and $R_{P2}$ are protective resistors, which are utilized to protect the charge meter. **h** Charging and discharging curves of $C_{test}$. Here, $C_{test}$ is 1.025 nF. **i** The simulated electric field result at electrode edge with different output voltage. The output voltage and charge of DC-TENG **j** w/o insulator and **k** w/ insulator. Inset figures of **j** and **k** are the photos of spark discharge of 3rd BD (Scale bar: 5 mm). Source data are provided as a Source data file.

It is noted that power sources do work only when they are under load rather than under short-circuit condition, and an important indicator of power sources is the external output voltage. In other words, achieving high output charge/current at high voltage is important for evaluating their output performance. We designed a circuit to measure the output charge of DC-TENG under different output voltage ($V_{CCE-FE}$) (Fig. 2g). When switch 1 ($S_1$) is off and switch 2 ($S_2$) is on, $V_{CCE-FE}$ gradually increases as the motion cycle increases, because output charges of DC-TENG gradually store in the testing capacitor ($C_{test}$). When $S_2$ is off and $S_1$ is on, charges stored in the $C_{test}$ are gradually released. From Fig. 2h, the charges stored in the $C_{test}$ should be equal to the charges released from the $C_{test,}$ indicating that the test result is not affected by leakage current of the test circuit, which is important for accurate test of output voltage (Supplementary Fig. 12 and Supplementary Note 5). For the normal DC-TENG (w/o insulator), output charge decreases from 174 nC to 45 nC with an increase of $V_{CCE-FE}$ from 0 V to 840 V (Fig. 2j), because electric field in 2nd BD easily reaches to the breakdown threshold with the $V_{CCE-FE}$ increase (Fig. 2i) and more charges are lost through the electrostatic breakdown in the 2nd BD. When the insulator is pasted on the edge of FE, the output charge of DC-TENG (w/ insulator) still maintains stable even the output voltage up to 1600 V, (Fig. 2k). This is attributed to the self-regulation ability of SEREF, which can accumulate electrostatic charges on the surface of the insulator to build the reverse electric field and can avoid the recurrence of electrostatic breakdown at 2nd BD. When $V_{CCE-FE}$ increases to the threshold voltage of 3rd BD, the electrostatic breakdown in 3rd BD occurs and the charges stored in $C_3$ (the equivalent capacitors formed by CCE and FE) will be released (Fig. 2e<i> and Fig. 2e<ii>). Therefore, limited by the 3rd BD, the maximum output voltage of DC-TENG with the insulator is a fixed value (Fig. 2k, Supplementary Figs. 12 and 13 and Supplementary Note 5), so it can be regarded as the open-circuit voltage ($V_{OC}$). It is noteworthy that the insulator can increase the $V_{OC}$ of DC-TENG because the spark discharge of 3rd BD must cross the insulator (the illustration of Fig. 2k).

## Performance enhancement of DC-TENG by SEREF

The DC-TENG with SEREF exhibits comprehensively enhanced performance and good reliability (Supplementary Fig. 14). In this strategy, the insulator not only decreases the charge loss in 2nd BD to enhance output charges in 1st BD, but also increases the $V_{OC}$ to a higher value with the increasing of the insulator's width (Supplementary Fig. 15). More importantly, the insulator's width (thickness) has nearly no impact on $Q_{SC}$ and $I_{SC}$ of DC-TENG (Fig. 3a, b and Supplementary Fig. 15) and greatly improves its output power, which indicates SEREF is the essential factor for performance improvement of DC-TENG again (Fig. 3c and Supplementary Fig. 16). In contrast, although $V_{OC}$ of the normal DC-TENG (w/o insulator) can increase with the gap between CCE and FE ($d_{CCE-FE}$) increase (Fig. 3b), but $Q_{SC}$ and $I_{SC}$ decreases (Fig. 3a) due to the enhancement of electrostatic breakdown in 2nd BD (Supplementary Fig. 17), producing a limited output power (Fig. 3c and Supplementary Fig. 16).

Compared to the performance of DC-TENG without insulator (the gap of CCE and FE: 0.5 mm) and that of DC-TENG with insulator, $Q_{SC}$ keeps stable, and $V_{OC}$ increases by 2.2 times, 4.2 times, and 5.9 times, and finally the average power has increased by 5.7 times, 7.9 times, and 10.7 times, respectively (Fig. 3a–c). In addition, similar experimental phenomena arise during the testing of DC-TENG's output energy as well (Supplementary Figs. 18 and 19 and Supplementary Notes 6, 7). It is noteworthy that the multiple of power increase is much higher than the multiple of voltage increase ($Q_{SC}$ keeps stable.), which cannot be explained by the conventional cognition-the output power is proportional to the product of $Q_{SC}$ and $V_{OC}$ (Supplementary Fig. 20 and Supplementary Note 8). Therefore, we speculate that there is still a critical but always neglected factor, which greatly affects the output performance of DC-TENG.

## Introducing Coulombic efficiency as a figure-of-merit for accurately evaluating the performance of TENG

In general, to evaluate and compare the performance of TENGs, the standards for quantifying the performance of TENGs were established by the cycle for maximized energy output (CMEO)[23–25]. The CMEO's $V$-$Q$ curve describes the relationship of the SCD of triboelectric layer ($\sigma_0$), ideal $V_{OC}$ ($V_{OC, ideal}$), and the inherent capacitor of TENG ($C_T$):

$$V_{OC, ideal} = \sigma_0 \times S/C_T \qquad (1)$$

where $S$ is the effective area of triboelectric layer, the reciprocal of absolute value of the $V$-$Q$ curve slope is the $C_T$. Ideally, the output energy at the fixed voltage $V_n$ ($n = 1, 2, 3$) can be obtained by multiplying the horizontal and vertical coordinates of the intersection between the voltage curve ($V = V_n$) and the CMEO's $V$-$Q$ curve (Fig. 3d, the shaded area represents the output energy.). Actually, considering the existence of electrostatic breakdown, the increase in output voltage will enhance the electric field strength around the electrode edge, leading to electrostatic breakdown, and then the SCD of triboelectric layer will decrease. This means that the $V$-$Q$ curve drawn by formula (1) is not static, and it is a dynamic curve with the increase of TENG's output voltage. We assume that, when the output voltage increases to $V_1$, the SCD of triboelectric layer will decrease to $\sigma_1$. It is noteworthy that $C_T$ is not affected by electrostatic breakdown. Therefore, substitute $\sigma_1$ into formula (1), a new $V$-$Q$ curve is obtained (the green dashed line in Fig. 3e), and the intersection of the voltage curve ($V = V_1$) and the new $V$-$Q$ curve shift left. Obviously, the actual output charge ($Q'_1$) will be lower than the ideal output charge ($Q_1$) (Fig. 3e). When the output voltage further increases to $V_2$ and $V_3$, SCD will decrease to $\sigma_2$ and $\sigma_3$, respectively, and the intersection continuously shifts left. In addition, when the output voltage increases to $V_4$, and $V_4$ is equal to $\sigma_4 \times S/C_T$, the output charge will decrease to zero. By connecting these intersections, a new curve for real energy output (CREO) can be obtained, and the product of the horizontal and vertical coordinates of each point on the curve represents the actual output energy of TENG (Fig. 3e, Supplementary Fig. 21 and Supplementary Note 9). Obviously, the difference between CREO and CMEO is caused by electrostatic breakdown, that's also why the actual energy output is much lower than the energy calculated by CMEO. From this perspective, our strategy utilizes the SEREF on insulator to suppress the electrostatic breakdown, which reduces the difference between CMEO and CREO, and makes CREO continuously close to CMEO to enhance the actual energy output (Fig. 3f).

To clearly understand the relationship of output energy, $Q_{SC}$, and $V_{OC}$, the $Q$-$V$ curve of TENG with voltage as the horizontal axis and charge as the vertical axis is plotted (Fig. 3g, Supplementary Fig. 22 and Supplementary Note 5), in which the output charge ($Q(V)$) as a function of the output voltage $V$:

$$Q(V) = (-a \times V + 1) \times Q_{SC} \qquad (2)$$

Here, the ratio of $Q(V)$ to $Q_{SC}$ is defined as Coulombic efficiency $\eta(V)$ to represent the charge utilization ratio at a fixed voltage:

$$\eta(V) = Q(V)/Q_{SC} = (-a \times V + 1) \qquad (3)$$

Here, $a$ is a constant, which is influenced by TENG's inherent capacitor, TENG's parasitic capacitor, and electrostatic breakdown (Supplementary Note 10). The larger $a$ is, the lower $\eta(V)$ is. Therefore, the output energy can be calculated as:

$$E(V) = \eta(V) \times Q_{SC} \times V \qquad (4)$$

According to formula (4) and Supplementary Fig. 23, even if TENGs with the same $Q_{SC}$ and $V_{OC}$, the maximum output energy is

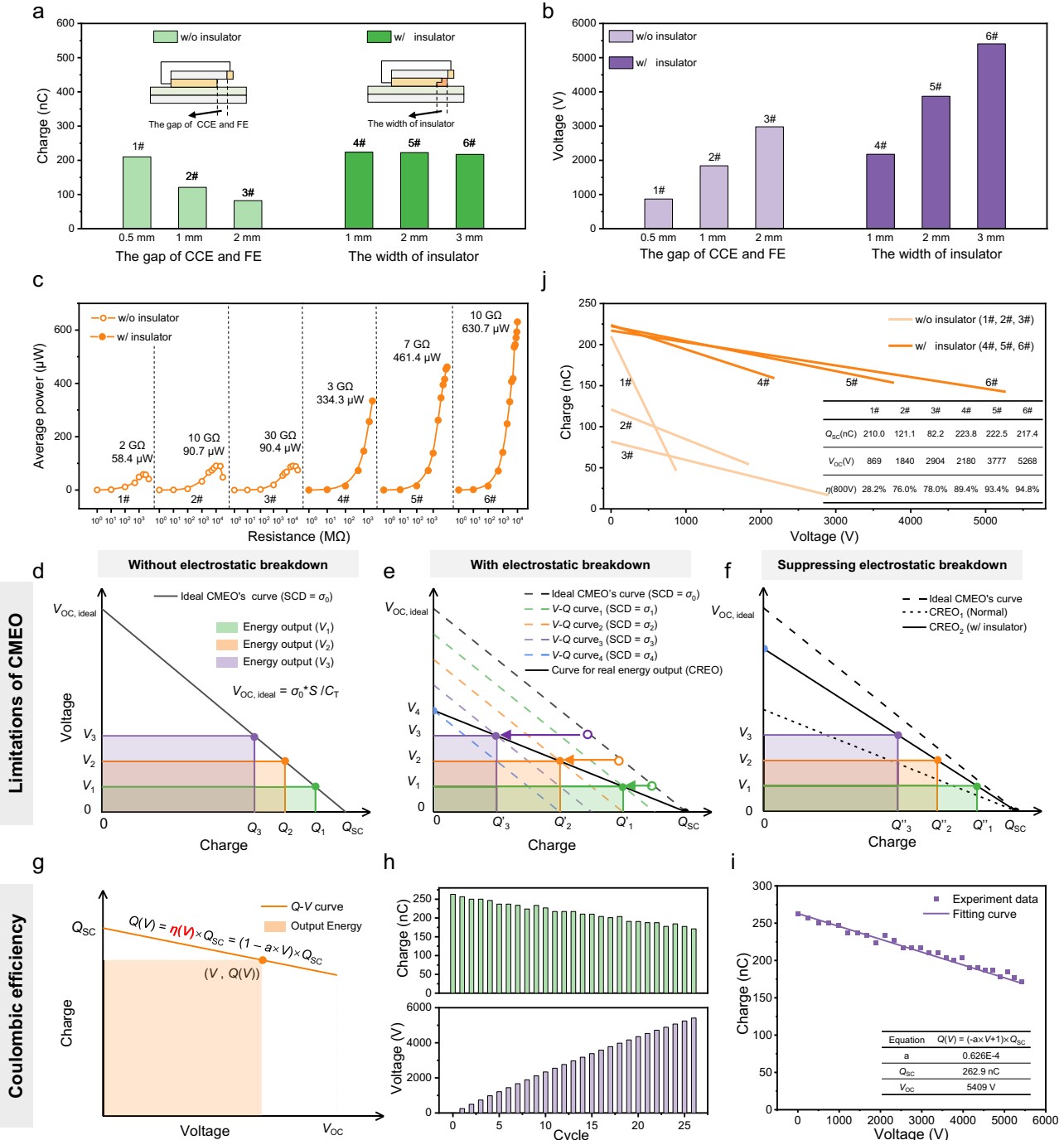

**Fig. 3 | Performance enhancement of DC-TENG by spontaneously established reverse electric field. a** $Q_{SC}$, **b** $V_{OC}$, and **c** average output power of DC-TENGs. **d** The ideal CMEO of DC-TENG. $Q_1$, $Q_2$, and $Q_3$ are the output charge when the output voltage of DC-TENG is $V_1$, $V_2$, and $V_3$, respectively. The shaded area represents the output energy. **e** The curve for real energy output (CREO). $Q'_1$, $Q'_2$, and $Q'_3$ are the actual output charge when the output voltage of DC-TENG are $V_1$, $V_2$, and $V_3$, respectively. **f** The CREO of DC-TENG w/ insulator. **g** The $Q$-$V$ curve of DC-TENG, where voltage as the horizontal axis and charge as the vertical axis. **h** The output charge and the output voltage after the nth motion cycle. **i** The $Q$-$V$ curve obtained by linearly fitting the experimental data in **h**. **j** The $Q$-$V$ curve of DC-TENGs with different parameters. Inset table shows Coulombic efficiency of different devices at the voltage of 800 V. Source data are provided as a Source data file.

different due to differences in $\eta(V)$ (Supplementary Note 10). Based on above analysis, we consider that $\eta(V)$ should be a parameter that is as important as $Q_{SC}$ or $V_{OC}$, for evaluating the output power of TENG.

The testing circuit in Fig. 2g <i> is utilized to obtain the $Q$-$V$ curve and $\eta(V)$. The Coulombic meter measures the total output charge ($\sum_{i=1}^{n} Q_n$) of DC-TENG, where $Q_n$ represents the output charge in the nth motion cycle (Fig. 3h). The output voltage after the

nth motion cycle ($V_n$) can be calculated by $C_{test}$ (Supplementary Fig. 24):

$$V_n = \frac{\sum_{i=1}^{n} Q_n}{C_{test}} \tag{5}$$

By linearly fitting the experimental data ($V_n$, $Q_n$), the curve shown in Fig. 3i can be obtained. The results of DC-TENG with different

structures in Fig. 3j demonstrate that $\eta(V)$ will be much higher than that of DC-TENG without insulator under the same voltage (Supplementary Fig. 25 and Supplementary Table 1). In addition, we calculated the maximum energy output of TENG by Coulombic efficiency, and compared them with the maximum output energy tested in the experiment. The comparison results are shown in the Supplementary Fig. 26. After introducing Coulombic efficiency, the calculated results are closer to the experimental results, which also demonstrate our strategy improving the performance of DC-TENG from three aspects: the output charge, the output voltage and Coulombic efficiency.

### Performance enhancement of different mode DC-TENGs by improving Coulombic efficiency

Currently, several mode DC-TENGs, including coplanar-electrode DC-TENG[31] (CDC-TENG), double dielectric layer DC-TENG[32] (DEDC-TENG), and rotary-mode DC-TENG[26], have been developed for different application scenarios. Due to the similar physical models and working principles, we speculate that the SEREF and $\eta(V)$ can also be utilized to improve their performance and evaluate their properties.

The principle of CDC-TENG is similar to the DC-TENG, which collects charges generated by electrostatic breakdown around electrodes (Fig. 4a, b). The difference is that two electrodes respectively act as CCEs to collect charges during the periodic motion (Supplementary Fig. 27a). The simulation results indicate that the breakdown domains also exist in the electrode edges (Supplementary Fig. 27b). Therefore, $V_{OC}$ increases with the gap of electrodes increase from 0.5 mm to 3.0 mm while $Q_{SC}$ rapidly decreases, and the output power remains basically unchanged (Fig. 4c, d). Because the enhanced electrostatic breakdown around electrodes results in a large charge loss, i.e., the low $\eta(V)$. When an insulator is pasted at the electrode edge, $Q_{SC}$, $V_{OC}$, and $\eta(V)$ of CDC-TENG are significantly improved (Supplementary Table 2), thus the power density increases by 5.8 times.

The DEDC-TENG utilizes electrostatic breakdown that occurs between CCE and TL to generate electricity and the leakage current of dielectric enhancement layer to maintain a continuous DC output, which is slightly different from that of DC-TENG (Fig. 4e and Supplementary Fig. 28a, b). Therefore, nitrile is often used as the dielectric enhancement layer due to the poor insulation performance (Supplementary Fig. 28c). Interestingly, in the process of DEDC-TENG manufacturing, nitrile always wraps around the edge of FE, which also can utilize side-discharge to accumulate charges on its side surface and generate a reverse electric field to suppress electrostatic breakdown. However, due to the poor insulation performance of nitrile, the negative charges accumulated on its side surface due to electrostatic breakdown will migrate inward (Fig. 4e) and offset the positive charges obtained by triboelectrification, which reduces suppression effect of SEREF on side-discharge. When polyimide (PI, a material with strong insulation performance) is pasted on the FE edge (Fig. 4f), the performance of DEDC-TENG ($Q_{SC}$, $V_{OC}$, and $\eta(V)$) is also greatly improved (Fig. 4g, Supplementary Fig. 28d and Supplementary Table 3). In addition, the experimental results indicate that materials with stronger insulation performance are more suitable for establishing SEREF, because it can prevent charge migrate to the electrode effectively and accumulate charge on insulator's surface quickly.

The rotary-mode DC-TENG can increase output current by integrating multiple units on a device (Fig. 4h), and achieve constant output current when the device rotates at a constant speed (Fig. 4i). When $K$ is 3 ($K$ represents the number of units) (Supplementary Fig. 29a), the $I_{SC}$ and maximum power ($P_{max}$) of DC-TENG with the insulator increases fourfold (Fig. 4i) and 30-fold (Supplementary Fig. 29b) compared with DC-TENG without the insulator, indicating the universality of our strategy. In addition, according to the $I$-$V$ curve ($I$-$V$ curve represents the output current as a function of the output voltage $I(V)$.) (Fig. 4j and Supplementary Fig. 30), the maximum output power ($P_{max}$) of DC-TENG without and with insulator ($K$ = 3) can be achieved

when the output voltage is $V_{\eta = 58.4\%}$ (output voltage corresponding to $\eta(V)$ of 58.4%) and $V_{OC}$, respectively (Supplementary Fig. 29c), which is consistent with previous conclusion (Supplementary Note 10). Remarkably, the $I_{SC}$ of DC-TENG ($K$ = 9) increases to 3.10 μA (Fig. 4i and Supplementary Fig. 31) and it further achieves a record-high power density up to 2.32 W m$^{-2}$ Hz$^{-1}$ (Fig. 4k and Supplementary Table 4), which is 54-fold of the previous DC-TENG and 16-fold of the integration design of micro-electrode DC-TENG (MDC-TENG) (Supplementary Fig. 32, Supplementary Table 5 and Supplementary Note 11).

### Performance enhancement of AC-TENG by improving Coulombic efficiency

Beyond DC-TENGs based on triboelectrification and electrostatic breakdown, the conventional AC-TENGs based on triboelectrification and electrostatic induction also suffer from the serious side-discharge problem (Fig. 5a, Supplementary Figs. 33–35 and Supplementary Note 12). When the gap of electrodes increases from 0.5 mm to 3.0 mm (Fig. 5b <i> and <ii>), $Q_{SC}$ rapidly decreases (Fig. 5c), but the output power remains stable (Fig. 5d). The simulated results indicate that electrostatic breakdown at the edge of the electrode also exists in AC-TENG (Supplementary Fig. 33a), which will decrease SCD on the triboelectric layer. Then, the ideal CMEO's curve is no longer suitable for directly calculating actual energy output of AC-TENG (CMEO of AC-TENG is a parallelogram due to alternating signal.) (Fig. 5e, f). When the output voltage is $V_1$, the actual output energy (the shaded area in Fig. 5f) and output charge ($Q'_1$) are lower than ideal output energy (the shaded area in Fig. 5e) and output charge ($Q_1$). Accordingly, $\eta(V)$ can also be considered as a figure-of-merit for evaluating the performance of AC-TENG. Figure 5g and Supplementary Table 6 demonstrate that the strategy of SEREF on insulators is also suitable for AC-TENG (Fig. 5b <iii>), which can increase Coulombic efficiency, acquiring a higher output power.

More importantly, relying on suitable triboelectric material pairs to enhance triboelectrification (Supplementary Fig. 33e) and combining this proposed strategy to suppress side-discharge and increase Coulombic efficiency (Fig. 5h and Supplementary Table 7), the AC-TENG with the common triboelectric material pairs of Polytetrafluoroethylene (PTFE) and Nylon (PA) can produce power density up to 6.15 W m$^{-2}$ Hz$^{-1}$ (Supplementary Fig. 33f, g, the output charge density is 0.69 mC m$^{-2}$), which is the highest value reported for sliding mode AC-TENGs[33–37] (Fig. 5i). Overall, these results demonstrate the strategy is universal and effective for performance improvement of various mode TENGs.

## Discussion

Here, we propose a universal and effective strategy to solve the troublesome side-discharge issue in TENGs by the spontaneously established reverse electric field, which is carried out only by pasting an insulator at the electrode edge to accumulate static charges and to establish a reverse electric field for suppressing electrostatic breakdown. Then, it is demonstrated that our strategy not only can improve short-circuit charge ($Q_{SC}$) and open-circuit voltage ($V_{OC}$) of TENGs like other strategies (increasing electrostatic breakdown threshold), but also can improve the performance of TENGs under load conditions by improving Coulombic efficiency. With verified devices, the enhanced average power density of 2.3 W m$^{-2}$ Hz$^{-1}$ (increased by 54 times) in DC-TENG and a record-breaking average power density of 6.15 W m$^{-2}$ Hz$^{-1}$ in AC-TENG (increased by 22 times) strongly demonstrate the universality and effectiveness of the strategy. It is noteworthy that our strategy for modulating the electric field intensity to suppress electrostatic breakdown in the breakdown domain has not been previously reported in the research field of TENG. Furthermore, we hypothesize that in addition to utilizing surface charge for electric field regulation, adjusting the terminal voltage of the electrode may also serve as a viable method (Supplementary Fig. 36 and Supplementary Note 13). More importantly, the performance of TENGs could be further

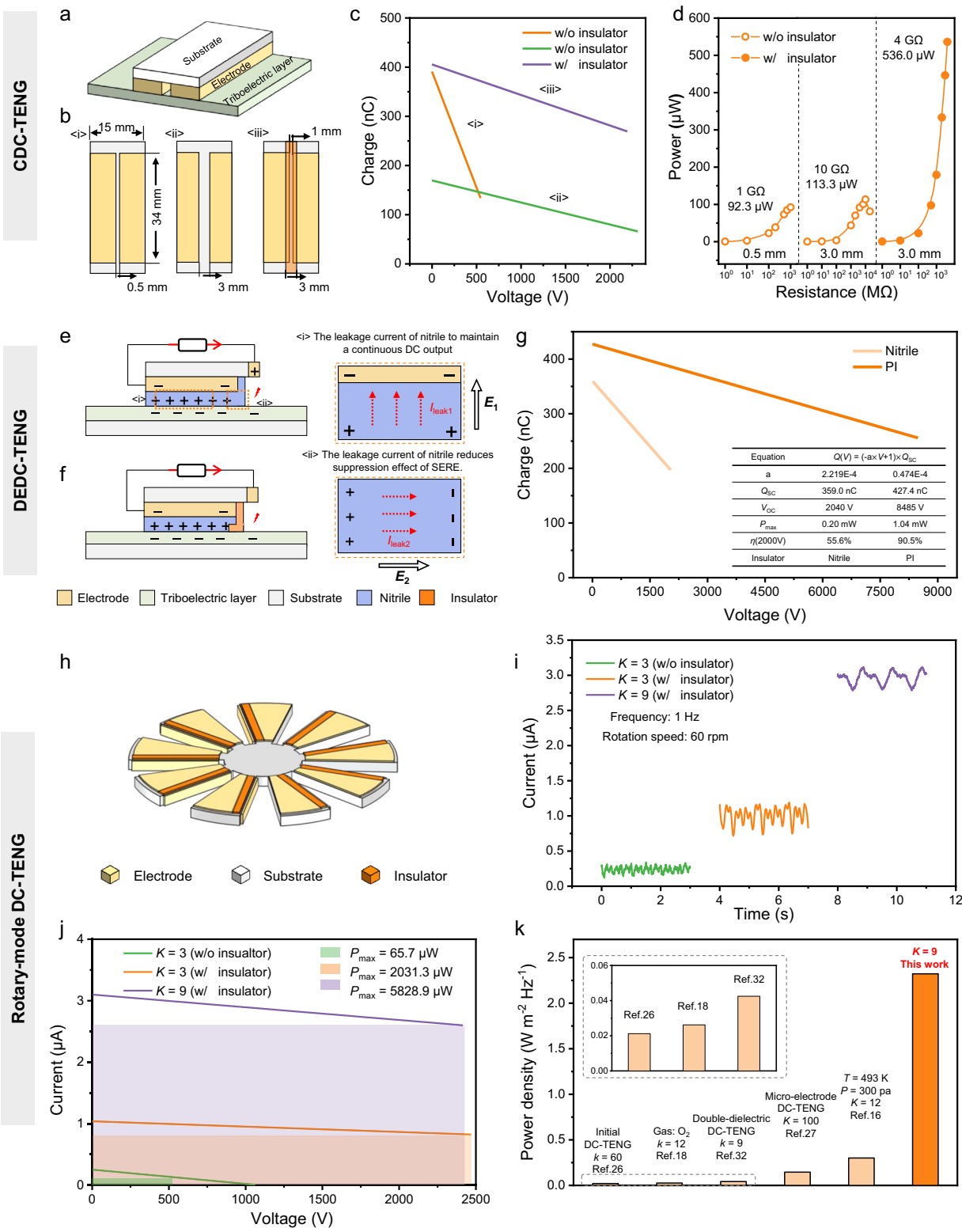

**Fig. 4 | Performance enhancement of different mode DC-TENGs by improving Coulombic efficiency. a** The structure of CDC-TENG. **b** <i> CDC-TENG without insulator, the gap of electrodes is 0.5 mm; <ii> CDC-TENG w/o insulator, the gap of electrodes is 3.0 mm; <iii> CDC-TENG w/o insulator, the gap of electrodes is 1 mm, and the width of insulator is 3.0 mm. **c** The *Q-V* curve of CDC-TENG. **d** The output power of CDC-TENG. **e** The structure diagram of DEDC-TENG, nitrile is used to enhance triboelectrification. **f** The structure diagram of DEDC-TENG w/ insulator.

**g** The *Q-V* curve obtained by linearly fitting the experimental data. **h** The structure diagram of rotary-mode DC-TENG (*K* = 9). **i** The constant current of DC-TENG (The frequency is 1 Hz, and the rotation speed is 60 rpm). **j** The *I-V* curve of DC-TENG w/ or w/o insulator. **k** The represented average power density of rotary mode DC-TENG with different optimization strategies. Source data are provided as a Source data file.

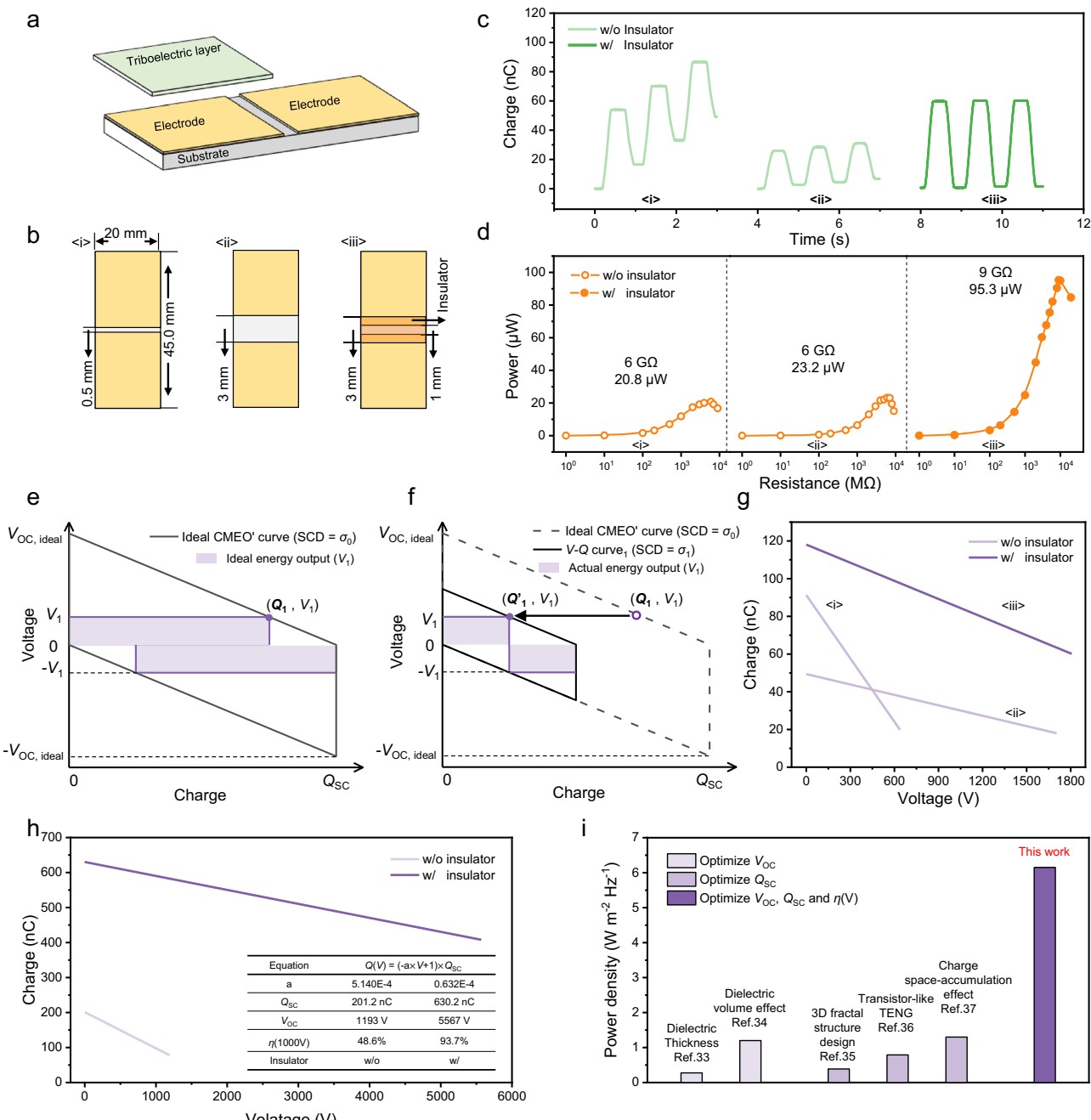

**Fig. 5 | Performance enhancement of AC-TENG by improving Coulombic efficiency. a** The structure of AC-TENG. **b** <i> AC-TENG w/o insulator, the gap of electrodes is 0.5 mm; <ii> AC-TENG w/o insulator, the gap of electrodes is 3.0 mm; <iii> AC-TENG w/o insulator, the gap of electrodes is 1 mm, and the width of insulator is 3.0 mm. **c** The output charge of AC-TENG. **d** The output power of AC-TENG. **e** The ideal CMEO's curve of AC-TENG. **f** The actual $V$-$Q$ curve with SCD of $\sigma_1$. **g** The $Q$-$V$ curve of AC-TENG w/ and w/o insulator. **h** The $Q$-$V$ curve of AC-TENG with double dielectric layers. **i** The represented average power density of AC-TENG with different optimization strategies. Source data are provided as a Source data file.

optimized by the synergetic enhancement of improved triboelectric charge density, as many previous works reported, and improved Coulombic efficiency as our strategy in the future.

It is noteworthy that the troublesome side-discharge problem widely exists in sliding mode AC/DC-TENGs with high SCD and limits the highest achievable output power density of TENGs. The SEREF on insulators presented in this work provides a promising solution to substantially enhance the output performance of TENGs. In addition, a new figure-of-merit, Coulombic efficiency, is proposed and demonstrated for correctly quantifying the output performance of TENGs, overcoming the issue of SCD decline dynamically

caused by electrostatic breakdown. More importantly, Coulombic efficiency provides a clear direction for guiding the design of high-performance TENGs and could be a standardized parameter for quantifying the performance of TENGs. Moreover, the $Q$-$V$ (or $I$-$V$) curve of TENGs drawn by Coulombic efficiency, $V_{OC}$, and $Q_{SC}$ can intuitively determine the performance of TENGs and provide the most direct data reference for subsequent power management circuit design. Overall, the strategy for improving performance and methods for evaluating the performance of TENG provided here set the foundation for the further applications and industrialization of TENG technology.

## Methods

### Fabrication of the sliding mode DC-TENG, CDC-TENG and DEDC-TENG

The substrates acrylic block (40 mm × 15 mm × 5 mm) of DC-TENG, CDC-TENG, and DEDC-TENG were cut using a laser cutter (PLS6.75, Universal Laser System). DC-TENG was composed of FE (35 mm × 14.5/14/13 mm × 0.05 mm) and CCE (35 mm × 0.05 mm × 5 mm), which were fabricated by the copper foil. Polyimide (PI) was used as the insulator and pasted on the edge of FE. The detailed structural diagrams of DC-TENG are shown in Supplementary Fig. 15a. The structure of DEDC-TENG was similar with DC-TENG. The only difference was that nitrile (35 mm × 15 mm × 5 mm) was pasted on the FE (35 mm × 14 mm × 5 mm). CDC-TENG was composed of two electrodes (35 mm × 7.25/7/6 mm × 0.05 mm), which were attached on the bottom surface of acrylic block, with a horizontal distance of 0.5 mm, 1.0 mm, and 3.0 mm. PI was used as the insulator and pasted on the edge of electrode. The detailed structural diagrams of CDC-TENG are shown in the Fig. 4b.

Three triboelectric layers (TLs) with different materials were made by adhering a foam layer on acrylic substrates, followed by a layer of fluorinated ethylene propylene (FEP), polyvinyl chloride (PVC), or ethylene-terafluoroethlene (ETFE) (8.0 cm × 5.0 cm × 50 μm), respectively. The TLs used in Figs. 2, and 3a–c, j were PVC, that in Figs. 3h, i and 4c, d, g were ETFE, and that in Supplementary Fig. 7a was FEP.

### Fabrication of the rotary mode DC-TENG

The substrate acrylic block of the rotary mode DC-TENG (diameter: 64 mm) was cut using a laser cutter. Then, FE and CCE were fabricated with copper foil and pasted on the substrate acrylic block. PI was used as the insulator and pasted on the edge of FE. The detailed structural diagram of the rotary mode DC-TENG ($K = 9$) is shown in Supplementary Fig. 31. The TL was made by adhering a foam layer on acrylic substrates, followed by a layer of ETFE film.

### Fabrication of the sliding mode AC-TENG

Two electrodes (20 mm × 20 mm × 0.05 mm) were attached on the bottom surface of acrylic block (45 mm × 20 mm × 5 mm), with a horizontal distance of 0.5 mm, 1.0 mm and 3.0 mm. PI was used as the insulator and pasted on the edge of electrode. The TL was made by adhering a foam layer on acrylic substrates, followed by a layer of PI film (20 mm × 20 mm × 0.05 mm).

### Measurement and characterization

The output current and output charge were measured by an electrostatic electrometer (Keithley 6514). Relative sliding motions of devices were achieved by a linear motor (TSMV120-1S, LinMot). The detailed sliding motion parameters are shown in Supplementary Table 8. The rotary process of rotary mode DC-TENG was achieved by a commercial programmable stepper motor (86BYG250D). Photos in this paper were taken with a digital camera (Nikon D750). FEM simulation was carried out in COMSOL Multiphysics software. The detailed simulation parameters are shown in the Supplementary Table 9.

## Data availability

The data that support the findings of this study are available from the corresponding author upon request. Source data are provided with this paper.

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

## Acknowledgements
Research was supported by the National Key R & D Project from Minister of Science and Technology (2021YFA1201602), National Natural Science Foundation of China (Grant no. 62204017, U21A20147 and 22109013), China Postdoctoral Science Foundation (2021M703172), Innovation Project of Ocean Science and Technology (22-3-3-hygg-18-hy), and the Fundamental Research Funds for the Central Universities (E1E46802).

## Author contributions
Y.G., D.L., and J.W. conceived the idea and wrote and revised the manuscript. Y.G. designed and fabricated the devices. Y.G. and D.L. performed the experiments and data measurements. Y.G., D.L., L.H., J.W., and Z.L.W. analyzed the data. J.Z. and L.Z. helped with the experiments. D.L., J.W., and Z.L.W. supervised this work. All the authors discussed the results and commented on the manuscript.

## Competing interests
The authors declare no competing interests.
