## [Peer Review File · Nature Communications]

REVIEWER COMMENTS

Reviewer #1 (Remarks to the Author):

The manuscript reported a strategy to enhance the performance of TENGs by spontaneously established reverse electric field. It is a continuous work of a previous study (Nat Commun 14, 3218 (2023). <https://doi.org/10.1038/s41467-023-38815-9>) reported by the same group. Mechanisms reported here are similar to what was reported in the reported one.

- 1) The unit " $W m^{-2} Hz^{-1}$ " lacks well-defined physical meaning.
- 2) The max output power at a load of over several $G\Omega$ due to discharge was not an optimised parameter to characterize the TENG. Based on our experience, the power could be even higher if one has a higher load. Instead, the energy output would be a better parameter for comparison.
- 3) Although the performance has been improved by the strategy, the energy output was not impressive. Figure 2h shows an energy output of about 200 nC/s.
- 4) It is strange to use Hz for the rotary system presented in Figure 4. Why not use rpm?

Reviewer #2 (Remarks to the Author):

This work by Gao et al., introduces a new concept on how to reduce the side-discharge issues related to sliding mode TENG, by introducing a reverse electric field. They highlight the parameter, "Coulombic-efficiency" as a method of estimating the performance of TENG. The TENG with the reverse electric field effect, obtained by fabricating an insulator layer on the electrode, demonstrate significant enhancement in output power generation. This an exciting paper, and can be recommended for publication in Nature Communications once the following queries are satisfactorily addressed.

1. Author should provide clarifications on how charges move from FE (Friction electrode) to charge collecting electrode (CCE). The potential difference graph is given in the paper; however, a suitable description should also be included to further improve the clarity. Authors can refer to a similar study <https://doi.org/10.1021/acsnano.0c00138> for example, which contains such a description.
2. When the FE move from left to right there will be negative triboelectric charges on the TL (triboelectric layer) due to triboelectric effect which can create a negative potential on TL, hence a potential difference between FE and CCE. In figure 2b, author has mentioned that there is no charge transfer between FE and CCE when sliding from right to left. Therefore, authors need to elaborate on how the charges transfer between FE and CCE for a whole unit cycle (right to left and left to right)?

3. In Figure 3, authors have mentioned that Fig. 3b represents the Voc. If the Voc was measured between FE and CCE, this means that there is (theoretically) no charge transfer from FE to CCE. However, the graph shows that the Voc increases with the increase of this gap between FE and CCE. Can the authors explain the reasons behind this?

4. The authors mentioned that the average power of the “DC-TENG with insulator” increased by 5.7 times, 7.9 times, and 10.7 times when the insulator width was increased, whereas the voltage increased by 2.2 times, 4.2 times, and 5.9 times (while Qsc was kept stable). Can authors explain here so as to why the power output improvement is not proportional to the voltage increment?

5. Can authors explain the motion profiles they used for their experiments, so that the readers are aware of the test conditions? Please include the information about the rate of movement (amplitude, frequency/velocity) as well as the contact force if available.

Reviewer #3 (Remarks to the Author):

The manuscript presents a technological development in the field of triboelectric nanogenerator, which is a relevant topic for the scientific community.

The work is focused on the main characteristics for a long-term operation, which is a proper strategy since it is relevant for practical purposes. Deep understanding of the physical mechanism that limit the development of new technologies play a relevant role in basic science.

As a main backward of this technology is claimed that the discharge due to electrostatic breakdown inducing a loss of power density. It is in this aspect that I will focus my attention on the review of the results.

The analysis is not adequate since it is not correct the claim in the introduction where the authors are assuming that this effect is unexpected and indicating that it is limiting factor for practical power management.

High power management of the BEOL or interconnect technology is well-known in the semiconductor industry, and different techniques and proper design can be implemented to mitigate the early breakdown of the interconnect dielectrics. As an example, different automotive application implements 10kV signal in dielectrics of 10 -20 microns. Therefore, it is possible to overcome the challenges that are present in this manuscript as a limiting factor.

Moreover, the repetitive loss of charge reported in the manuscript should also be analyzed in terms of the reliability of the proposed devices. A deep understanding of the degradation mechanism during the process of discharge should also be provided.

Based on this analysis I suggest considering publication of the manuscript after a mayor review in terms of the physics behind the discharge of the storage charge considering the state of the art of interconnect technology. I recommend the inclusion of cross section TEM to confirm the dimension of the DUTs.

Point-by-point responses to the reviewers' comments

We (the authors) would like to thank the editor and reviewers for dedicating their valuable time and expertise to assess the manuscript. We appreciate the reviewers' thorough and insightful comments, which significantly contributed to enhancing the quality of the present work. In the following we elucidate our response to each remark, along with the corresponding revisions made in the manuscript, in our pursuit of continued consideration for this submission. (Comments in Red, responses in Black, revisions in Blue.)

Response to Reviewer #1

General remark

The manuscript reported a strategy to enhance the performance of TENGs by spontaneously established reverse electric field. It is a continuous work of a previous study (Nat Commun 14, 3218 (2023), <https://doi.org/10.1038/s41467-023-38815-9>) reported by the same group. Mechanisms reported here are similar to what was reported in the reported one.

Response

We are grateful for the reviewer's continuous attention of our group works, which is a great encouragement to us.

In 2019, our group firstly proposed the direct-current triboelectric nanogenerator (DC-TENG) based on triboelectrification and electrostatic breakdown (*Sci. Adv.* **5**, eaav6437 (2019)). By placing an electrode on the side of the substrate, a fixed breakdown domain can be constructed between the charge collection electrode (CCE) and the triboelectric layer (TL) for electrostatic breakdown, and then a DC signal can be generated.

Previously, we assumed that the output performance of DC-TENG is decided only by one discharge domain-between the CCE and TL. However, it is difficult to explain experimental results such as the output charge decay at large external loads. Recently, we comprehensively analyzed the experimental and simulation results for DC-TENG (*Nat. Commun.* **14**, 3218 (2023).), and demonstrate that there are three breakdown domains: the first breakdown domain (1st BD) between CCE and TL, the second breakdown domain (2nd BD) between friction electrode (FE) and TL, and the third breakdown domain (3rd BD) between CCE and FE (**Figure R1**). More importantly, we systematically imaged, defined, and regulated three discharge domains in DC-TENG, then a “cask model” was developed to bridge the cascaded-capacitor-breakdown dynamic model in the ideal condition and real outputs.

Although our previous work indicated that suppressing the electrostatic breakdown of 2nd BD can improve the performance of DC-TENG, there are still no research on the following aspects: **(1) how to suppress electrostatic breakdown; (2) the mechanism for suppressing electrostatic breakdown; (3) which parameters affect the effectiveness of suppressing electrostatic breakdown; (4) how to improve TENG's performance by suppressing electrostatic breakdown. We have answered the above questions in this manuscript.**

Figure R1 (Figure 2a). The schematic diagram of DC-TENG. There are three breakdown domains: 1st BD, 2nd BD and 3rd BD.

Comparing with the previous work, we report an important progress on these fundamental problems that the spontaneously established reverse electric field (SEREF) between the electrode and triboelectric layer restricts electrostatic breakdown by decreasing the electric field strength below critical breakdown electric field, which can be achieved by only pasting an insulator at the electrode edge. More importantly, the established electric field is self-regulation depending on the triboelectric charge density. Three unique capabilities of this strategy, which are not readily available in previous reports, have been demonstrated in this manuscript:

1. This strategy applies to both direct-current TENG (DC-TENG) based on electrostatic breakdown and alternate-current TENG (AC-TENG) based on electrostatic induction. With verified devices, the enhanced average power density of $2.30 \text{ W m}^{-2} \text{ Hz}^{-1}$ (increased by 54 times) in DC-TENG and a record-breaking average power density of $6.15 \text{ W m}^{-2} \text{ Hz}^{-1}$ (increased by 22 times) in AC-TENG strongly demonstrate the universality and effectiveness of the strategy.
2. This strategy is very simple to implement by just pasting a layer of insulation tape at the edge of the electrode, with the self-regulation behavior to adapt TENGs of different triboelectric charge density.
3. This strategy not only improves performance of TENG in the terms of Q_{SC} and V_{OC} like conventional strategies, but is also the first to improve performance from Coulombic efficiency.

Besides, a new figure-of-merit, Coulombic efficiency, is proposed and demonstrated for correctly quantifying the output performance of TENGs. Two unique characteristics of Coulombic efficiency compared with previous CMEO (the cycle for maximized energy output):

1. Coulomb efficiency proposed in this paper is not contradictory to CMEO. CMEO is more suitable for theoretical performance evaluation of TENG, and Coulomb efficiency also reflects the surface charge density decline degree dynamically caused by electrostatic breakdown, pointing out the direction for performance optimization of TENGs.
2. The Q - V (or I - V) curve of TENG drawn by Coulombic efficiency, V_{OC} , and Q_{SC} , as important as the I - V curve

for solar cells, can not only intuitively determine the performance of TENG, but also provide the most direct data reference for subsequent power management circuit design.

In addition to these we have reported in the manuscript, we also made the following supplements in the Supporting Information to further explain the difference in mechanism between the present work and the previous one.

Revisions

The revised part in the present manuscript is as follows:

We have made a revision of “Although our previous work indicated that suppressing the electrostatic breakdown of 2nd BD can improve the performance of DC-TENG (Supplementary Fig. 9)^{28, 29},” in the second paragraph of “SEREF for regulating breakdown domains of DC-TENG”.

The revised part in the present supplementary information is as follows:

Figure R2 (Added Supplementary Figure 9). Experimental verification of the principle of suppressing electrostatic breakdown of pasting an insulator at the edge of FE in DC-TENG. (a) The structure diagram of DC-TENG without insulator. The gap between CCE and FE is 0.5 mm. **(b)** The structure diagram of DC-TENG with insulator. There is still about 0.4 mm of air gap in the 2nd BD between CCE and FE. **(c)** The Q - V curve of DC-TENG. Previous work speculated that the insulator pasted at the edge of electrode was to completely replace the air in the 2nd BD, increasing the breakdown threshold of 2nd BD. Here, two experimental evidences indicate that it may be unreasonable. Firstly, when the insulator is nitrile, the effect of suppressing electrostatic breakdown is poor (Fig. 4e-g). Furthermore, even in the presence of a minor air gap within the 2nd BD, the effect of suppressing air breakdown still exists. These can be explained by the theory proposed in this work.

Remark-1

1) The unit " $\text{W m}^{-2} \text{Hz}^{-1}$ " lacks well-defined physical meaning.

Response

We appreciate your professional review and reasonable suggestion. Given that TENG converts the mechanical energy in the periodic mechanical motion into electricity, its output power depends on the device area of TENG and the motion frequency^{1,2}, so the power density is not an ideal parameter to assess the output capability of TENG. Energy density has been demonstrated as one of the most standardized parameters for TENG that is intrinsically not affected by the motion frequency. Here, the unit " $\text{W m}^{-2} \text{Hz}^{-1}$ " can also be expressed as the energy density, i.e., " J m^{-2} ". Given that the power density at different frequencies is often obtained firstly, the unit " $\text{W m}^{-2} \text{Hz}^{-1}$ " is widely used in many previous works for fair comparison. Therefore, we also use this unit to compare the output performance of our devices with the previous works. (*Nat. Commun.* **11**, 4277 (2020), *Energy Environ. Sci.* **16**, 3486-3496 (2023), *Adv. Mater.* **35**, 2302954 (2023)).

Remark-2

2) The max output power at a load of over several $\text{G}\Omega$ due to discharge was not an optimised parameter to characterize the TENG. Based on our experience, the power could be even higher if one has a higher load. Instead, the energy output would be a better parameter for comparison.

Response

We appreciate your professional review and reasonable suggestion. We will give a comprehensive analysis to show the similarity and difference of the maximum output power and maximum output energy. In our work, when measuring the optimal output power of TENG by changing the resistance, two possible results appeared as shown in **Figure R3**. One observation is that the output power reached its maximum value and then decreased as the resistance increases from several $\text{M}\Omega$ to several $\text{G}\Omega$ (**Figure R3a-c**), indicating a power inflection point; another observation is that the discharge signal occurs before the power inflection point (**Figure R3d-f**). In addition, other works about TENG's output power have also reported that the optimal resistance of TENG decreases with increasing motion frequency^{3,4}, which can also be verified by TENG's theoretical capacitance model⁵. Indeed, this poses difficulty in comparing the performance of different TENGs.

Figure R3 (Revised Supplementary Figure 16). The output power of DC-TENG with different structure parameters. (a)  The structure diagram of DC-TENG without insulator. <ii> The output power of DC-TENG. The gap between CCE and FE is 0.5 mm. (b)  The structure diagram of DC-TENG without insulator. <ii> The output power of DC-TENG. The gap between CCE and FE is 1.0 mm. (c)  The structure diagram of DC-TENG without insulator. <ii> The output power of DC-TENG. The gap between CCE and FE is 2.0 mm. (d)  The structure diagram of DC-TENG with insulator. The width of insulator is 1.0 mm. <ii> The output power and <iii> the output voltage (The resistance is 4 GΩ.) of DC-TENG. (e)  The structure diagram of DC-TENG with insulator. The width of insulator is 2.0 mm. <ii> The output power and <iii> the output voltage (The resistance is 7 GΩ.) of DC-TENG. (f)  The structure diagram of DC-TENG with insulator. The width of insulator is 3.0 mm. <ii> The output power and <iii> the output voltage (The resistance is 20 GΩ.) of DC-TENG. Obviously, when a breakdown signal occurs in the current signal, the corresponding output voltage approaches the threshold voltage of the 3rd BD (i.e. the open circuit voltage of DC-TENG).

The output energy of TENG for each motion cycle is not affected by the motion frequency. Here, the testing method for output voltage and $Q-V$ curve of TENG proposed in this work is shown in **Figure 2g**, which can also be utilized to measure the output energy of other TENGs (The schematic diagram of the testing circuit and the process of data processing are shown in **Figure R4**). The output energy of TENG can be calculated based on the formula ($E_T = 0.5 \times C_{test} \times (V_n^2 - V_{n-1}^2)$) (**Figure R4b-d** and **Figure R4f-g**). E_T is the energy generated by TENG in each motion cycle, and V_n is the voltage of C_{test} (the testing capacitor) after the n^{th} motion cycle. It is worth noting that when the voltage of C_{test} reaches the breakdown threshold voltage of the 3rd BD, discharge will occur and the energy stored in C_{test} will stop to increase (**Figure R4e** and **Figure R5**). **Figure R6** shows the output energy of the six devices proposed in **Figure R3**. For the devices with the power inflection point (the devices shown in **Figure R3a-c**), the highest output energy will be obtained at the intermediate motion cycle. For devices without the power inflection point (the devices shown in **Figure R3d-f**), the energy in the last cycle is often the highest.

Figure R4 (Added Supplementary Figure 18). The testing method for output energy of TENG. (a) The schematic diagram of the testing circuit. **** When the DC-TENG moves left, current flows from FE to CCE and charge is stored in the testing capacitor (C_{test}). **<ii>** When the DC-TENG moves right, current is zero. R_p is a protective resistor, which is utilized to protect the charge meter. **(b)-(d)** The schematic diagram of testing output energy for DC-TENG. The green/orange/purple shadow area represent the output energy of DC-TENG in the 1st/2nd/3rd motion cycle. Q_n is the total output charge after the n^{th} motion cycle. $Q_n - Q_{n-1}$ is the output charge of the n^{th} motion cycle. V_n is the output voltage after the n^{th} motion cycle. **(e)** The output charge and output voltage of DC-TENG (Without insulator, the gap between CCE and FE is 0.5 mm.). **(f)-(g)** The output energy of DC-TENG in the n^{th} motion cycle (Without insulator, the gap between CCE and FE is 0.5 mm.).

Figure R5 (Figure 2j-k). (a) The output voltage of DC-TENG without insulator. **(b)** The output voltage of DC-TENG with insulator. Inset figures of **(a)** and **(b)** are the photos of spark discharge of 3rd BD (Scale bar: 5 mm).

Figure R6 (Added Supplementary Figure 19). The output energy of DC-TENG. (a) The output energy of DC-TENG without insulator. <ii><iii> The gap between FE and CCE is 0.5 mm, 1.0 mm and 2.0 mm, respectively. The detailed structure diagram is shown in **Supplementary Fig. 16a-c**. **(b)** The output energy of DC-TENG with insulator. <ii><iii> The width of insulator is 1.0 mm, 2.0 mm and 3.0 mm, respectively. The detailed structure diagram is shown in **Supplementary Fig. 16d-f**.

The above experimental results indicate that there is a certain similarity between the maximum output power of TENG measured by varying resistances and the maximum output energy measured by capacitances. **We consider that the key point is not which method is more suitable for characterizing TENG's performance, but rather the specific conditions under which TENG can achieve its maximum output power/energy. This has been discussed in detail in this work.**

We utilized the Coulomb efficiency proposed in this work, coupled with open-circuit voltage (V_{oc}) and short-circuit charge (Q_{sc}), to plot the Q - V curve of TENG.

$$Q(V) = \eta(V) \times Q_{sc} \quad (1)$$

The Q - V curve represents the output charge when the output voltage is V . The product of the horizontal and vertical coordinates of any points on the Q - V curve represents the output energy ($E(V)$) of TENG when the voltage is V .

$$E(V) = \eta(V) \times Q_{sc} \times V \quad (2)$$

Therefore, if the $\eta(V_{oc}) \leq 50\%$, the maximum output energy can be calculated as:

$$E_{max} = 50\% \times Q_{sc} \times V_{\eta=50\%} \quad (3)$$

In other words, TENG's output energy is maximum when the output voltage is $V_{\eta=50\%}$ ($V_{\eta=50\%}$ refers to the voltage corresponding to a Coulomb efficiency of 50%).

When the $\eta(V_{OC}) \geq 50\%$, the maximum output energy can be calculated as:

$$E_{\max} = \eta_{V_{OC}} \times Q_{SC} \times V_{OC} \quad (4)$$

and TENG's output energy is maximum when the output voltage is close to V_{OC} .

As shown in **Figure R7** and **Table R1**, the $\eta(V_{OC})$ of the devices in **Figure R3a-c** (They are also the device 1#, device 2# and device 3# in **Figure 3a.**) is less than 50%, so there is a power/energy inflection point for these devices. The $\eta(V_{OC})$ of the devices in **Figure R3d-f** (They are also the device 4#, device 5# and device 6# in **Figure 3a.**) is greater than 50%, so there is no power/energy inflection point for these devices, leading to a higher output power or energy at the higher output voltage.

Figure R7 (Revised Supplementary Figure 25). The Q - V curve of DC-TENGs with different structure parameters obtained by linearly fitting the experimental data. (a) The Q - V curve of DC-TENG without insulator. (i)-(iii) The gap between FE and CCE is 0.5 mm, 1.0 mm and 2.0 mm, respectively. The detailed structure diagram is shown in **Supplementary Fig. 16a-c**. (b) The Q - V curve of DC-TENG with insulator. (i)-(iii) The width of insulator is 1.0 mm, 2.0 mm and 3.0 mm, respectively. The detailed structure diagram is shown in **Supplementary Fig. 16d-f**. The Coulombic efficiency of these devices is shown in **Supplementary Table 1**.

Table R1 (Revised Supplementary Table 1). The Coulombic efficiency of DC-TENG with different structure parameters

	a	$Q_{sc}(nC)$	$V_{oc}(V)$	$\eta(500V)$	$\eta(869V)$	$\eta(1840V)$	$\eta(2180V)$	$\eta(2904V)$	$\eta(3777V)$	$\eta(5268V)$
w/o insulator (0.5 mm)	8.968E-4	210.0	869	55.2%	22.1%	–	–	–	–	–
w/o insulator (1.0 mm)	3.002E-4	121.1	1840	85.0%	73.9%	44.8%	–	–	–	–
w/o insulator (2.0 mm)	2.747E-4	82.2	2904	86.3%	76.1%	49.4%	40.1%	20.2%	–	–
w/ insulator (1.0 mm)	1.325E-4	223.8	2180	93.4%	88.5%	75.6%	71.1%	–	–	–
w/ insulator (2.0 mm)	0.821E-4	222.5	3777	95.9%	92.9%	84.9%	82.1%	76.2%	69.0%	–
w/ insulator (3.0 mm)	0.651E-4	217.4	5268	96.7%	94.3%	90.2%	85.8%	81.1%	75.4%	65.7%

In summary, the Coulombic efficiency we improved in this manuscript indicates that both power density and energy density are suitable for characterizing TENG’s performance. For devices with the maximum output power inflection point, there is also a maximum output energy inflection point. The inflection point depends on the specific conditions under which TENG can achieve its maximum output power/energy. In comparison, testing the output energy of TENG can provide a more intuitive understanding of the relationship between TENG's output voltage and output energy. We have added **Supplementary Figure 18** and **Supplementary Note 6** in the supplementary information to introduce the method for measuring output energy of DC-TENG. In addition, we have added **Supplementary Figure 19** and **Supplementary Note 7** in the supplementary information to give a detailed analysis of maximum output power and output energy for TENG.

Revisions

The revised part in the present manuscript is as follows:

We have made a revision of “Compared to the performance of DC-TENG without insulator (the gap of CCE and FE: 0.5 mm) and that of DC-TENG with insulator, Q_{sc} keeps stable and V_{oc} increases by 2.2 times, 4.2 times, and 5.9 times, and finally the average power has increased by 5.7 times, 7.9 times, and 10.7 times, respectively (**Fig. 3a-c**). In addition, similar experimental phenomena arise during the testing of DC-TENG’s output energy as well (**Supplementary Fig. 18-19** and **Supplementary Note 6-7**)” in the last paragraph of “Performance enhancement of DC-TENG by SEREF”.

The revised part in the present supplementary information is as follows:

Supplementary Note 6 The testing method for output energy of TENG

The output energy of TENG for each motion cycle is not affected by the motion frequency. Here, the testing method for output voltage and Q - V curve of TENG proposed in this work is shown in **Fig. 2g**, which can also be utilized to measure the output energy of other TENGs (The schematic diagram of the testing circuit and the process of data processing are shown in **Supplementary Fig. 18**). The output energy of TENG can be calculated based on the formula ($E_T = 0.5 \times C_{\text{test}} \times (V_n^2 - V_{n-1}^2)$) (**Supplementary Fig. 18b-d** and **Supplementary Fig. 18f-g**). E_T is the energy generated by TENG in each motion cycle, and V_n is the voltage of C_{test} (the testing capacitor) after the n^{th} motion cycle. It is worth noting that when the voltage of C_{test} reaches the breakdown threshold voltage of the 3rd BD, discharge will occur and the energy stored in C_{test} will stop to increase (**Supplementary Fig. 18e** and **Fig. 2j-k**).

Supplementary Note7 Comparison output power and output energy of TENG

Supplementary Fig. 19 shows the output energy of the six devices proposed in **Fig. 3a**. For the devices with the power inflection point (the device 1#, 2#, 3#), the highest output energy will be obtained at the intermediate motion cycle. For devices without the power inflection point (the device 4#, 5#, 6#), the energy in the last cycle is often the highest. The above experimental results indicate that there is a certain similarity between the maximum output power of TENG measured by varying resistances and the maximum output energy measured by capacitances. We consider that the key point is not which method is more suitable for characterizing TENG's performance, but rather the specific conditions under which TENG can achieve its maximum output power/energy.

According to **Supplementary Note 10**, it can be inferred that, if the $\eta(V_{\text{OC}}) \leq 50\%$, TENG's output energy is maximum when the output voltage is $V_{\eta=50\%}$; if the $\eta(V_{\text{OC}}) > 50\%$, TENG's output energy is maximum when the output voltage is close to V_{OC} . As shown in **Supplementary Fig. 25** and **Supplementary Table 1**, the $\eta(V_{\text{OC}})$ of the device 1#, device 2# and device 3# in **Fig. 3a** is less than 50%, so there is a power/energy inflection point for these devices. The $\eta(V_{\text{OC}})$ of the device 4#, device 5# and device 6# in **Fig. 3a** is greater than 50%, so the higher the output voltage, the greater the output power or energy for these devices.

In summary, the Coulombic efficiency we improved in this manuscript indicates that both power density and energy density are suitable for characterizing TENG's performance. For devices with maximum output power inflection point, there is also a maximum output energy inflection point. The inflection point depends on the specific conditions under which TENG can achieve its maximum output power/energy. In comparison, testing the output energy of TENG can provide a more intuitive understanding of the relationship between TENG's output voltage and output energy.

Remark-3

3) Although the performance has been improved by the strategy, the energy output was not impressive. Figure 2h shows an energy output of about 200 nC/s.

Response

We appreciate your professional review and generous comments. To measure the output charge of DC-TENG under different output voltage ($V_{\text{CCE-FE}}$), we designed a circuit as shown in **Figure 2g** (The detailed discussions are shown in **Supplementary Note 5**). **Figure 2h** shows the charges stored in the C_{test} of **Figure 2g**. The DC-TENG used in this experiment is the device 6# in **Figure 3a**. Comparing the performance of device 1# and 6# in **Figure 3a**, it can be observed that even though the short-circuit charge is consistent (**Figure 3a**), the output power increases from 58.4 μW to 630.7 μW (**Figure 3c**) due to different open-circuit voltage values (**Figure 3b**) and different Coulombic efficiencies (**Supplementary Table 1**).

It is worth noting that our strategy improving the performance of DC-TENG from three aspects: the output charge, the output voltage and Coulombic efficiency. Although the charge density in our work is not the highest reported value, enhancing V_{OC} and Coulombic efficiency have enabled the highest output power density, demonstrating the effectiveness of our proposed strategy for performance optimization of TENG again.

For example, for the rotary mode DC-TENG in this work, the output charge density is 1.2 mC m^{-2} , while the highest output charge density of 8.8 mC m^{-2} has been reported in the microstructure-designed direct-current TENG (MDC-TENG)⁶. The output power density of DC-TENG designed by our strategy is 16-time of that of MDC-TENG (**Figure 4k** and **Supplementary Note 11**). For the sliding mode AC-TENG working in atmosphere conditions, the highest output charge density of 1.63 mC m^{-2} has been reported in the charge-space-accumulation sliding mode TENG (CAS-S-TENG)⁷, while the output charge density for the AC-TENG in this work is only 0.69 mC m^{-2} . However, the power density of our AC-TENG is up to $6.2 \text{ W m}^{-2} \text{ Hz}^{-1}$, which is 5.7-time of that of CAS-S-TENG. These results indicate that output charge density is not the only parameter for evaluating and optimizing TENG performance. Therefore, we believe that the comprehensive enhancement in output charge, output voltage and Coulombic efficiency holds the key to further optimizing TENG's performance in the future.

Remark-4

4) It is strange to use Hz for the rotary system presented in Figure 4. Why not use rpm?

Response

We appreciate your professional review and reasonable suggestion. Generally, it is more appropriate to use the unit "rpm" for rotary systems rather than "Hz". Given the widespread adsorption of the unit "Hz" by various groups in numerous previous works (*Nat. Commun.* **11**, 4277 (2020), *Adv. Energy Mater.* **12**, 2201454 (2022), *Adv. Energy Mater.* **10**, 2002920 (2020), *Energy Environ. Sci.* **16**, 3486-3496 (2023)), we employ this unit in our study to facilitate comparisons with the results of our peers (A frequency of 1 Hz signifies to one revolution per second.). For the convenience of reviewers and potential readers, we have added the unit of "rpm" as a reference and provided the detailed motion parameters of our devices. We added this part as **Supplementary Table 4** in the supplementary information, revised **Figure 4i** in manuscript and **Supplementary Fig. 29b** in Supplementary information. In addition, we also added the detailed motion parameters of our devices in **Supplementary Table 8**.

Revisions

The revised part in the present manuscript is as follows:

We have made a revision of “Remarkably, the I_{SC} of DC-TENG ($K = 9$) increases to $3.10 \mu\text{A}$ (Fig. 4i and Supplementary Fig. 20) and it further achieves a record-high power density up to $2.32 \text{ W m}^{-2} \text{ Hz}^{-1}$ (Fig. 4k and Supplementary Table 4),” in the last paragraph of “Performance enhancement of different mode DC-TENGs by improving Coulombic efficiency”.

The revised part in the present manuscript is as follows:

Figure R8 (Revised Figure 4i). The constant current of DC-TENG (The frequency is 1 Hz and the rotation speed is 60 rpm.).

The revised part in the present Supplementary information is as follows:

Figure R9 (Revised Supplementary Figure 18b). The output power of the rotary mode DC-TENG (The frequency is 1 Hz and the rotation speed is 60 rpm.).

Supplementary Table 4. The detailed parameters of different rotation mode DC-TENGs.

Device	Parameters	Rotation Speed (rpm)	Frequency (Hz)	Area (cm²)	Power density (W m⁻² Hz⁻¹)
	Ref. 26	600	10	314	0.01
	Ref. 18	200	3.3	22	0.02
	Ref. 32	300	5	38	0.04
	Ref. 27	300	5	15	0.15
	Ref. 16	60	1	47	0.30
	This work	60	1	25	2.3

Revised Supplementary Table 8. The detailed motion parameters of sliding-mode TENG

Device	Acceleration	Deceleration	Maximum rate	Frequency	Distance	Pressure
DC-TENG (Figure 2)	0.2 m s ⁻²	0.2 m s ⁻²	0.1 m s ⁻¹	0.5 Hz	50 mm	10 N
DC-TENG (Figure 2)	0.8 m s ⁻²	0.8 m s ⁻²	0.2 m s ⁻¹	1 Hz	50 mm	10 N
CDC-TENG (Figure 4)	0.8 m s ⁻²	0.8 m s ⁻²	0.2 m s ⁻¹	1 Hz	50 mm	10 N
DEDC-TENG (Figure 4)	0.2 m s ⁻²	0.2 m s ⁻²	0.1 m s ⁻¹	0.5 Hz	50 mm	10 N
AC-TENG (Figure 5)	0.4 m s ⁻²	0.4 m s ⁻²	0.1 m s ⁻¹	1 Hz	25 mm	10 N

Response to Reviewer #2

General remark

This work by Gao et al., introduces a new concept on how to reduce the side-discharge issues related to sliding mode TENG, by introducing a reverse electric field. They highlight the parameter, “Coulombic-efficiency” as a method of estimating the performance of TENG. The TENG with the reverse electric field effect, obtained by fabricating an insulator layer on the electrode, demonstrate significant enhancement in output power generation. This an exciting paper, and can be recommended for publication in Nature Communications once the following queries are satisfactorily addressed.

Response

We highly appreciate the reviewer for carefully reviewing our work, and thank your positive comments on our research work, which has encouraged us greatly.

Remark-1

1. Author should provide clarifications on how charges move from FE (Friction electrode) to charge collecting electrode (CCE). The potential difference graph is given in the paper; however, a suitable description should also be included to further improve the clarity. Authors can refer to a similar study <https://doi.org/10.1021/acsnano.0c00138> for example, which contains such a description.

Response

We appreciate your professional review and reasonable suggestion. We have provided a detailed description of the charge transfer mechanism in our manuscript and supplemented it with detailed schematic diagrams in the supporting information.

Revisions

The revised part in the present manuscript is as follows:

We have made a revision of “Based on the triboelectrification effect between the friction electrode (FE) and triboelectric layer (TL), negative charges and positive charges are generated on the surface of TL and FE (**Fig. 2a **). When DC-TENG moves left, a unidirectional electric field will be built between charge collection electrode (CCE) and TL to induce electrostatic breakdown, and negative charges transfer from the surface of TL to CCE driven by the Coulomb force; due to the significant potential difference between CCE and FE (**Fig. 2a <ii>** and **Supplementary Fig. 8a**), negative charges will transfer from CCE to FE, generating DC output in external circuit. If the DC-TENG continues to move toward the left, a continuous DC output can be obtained. When DC-TENG moves right, the electric field between CCE and TL cannot induce electrostatic breakdown because the surface charge of TL below CCE is nearly zero (**Fig. 2a <iii>**); thus, there is no charge transfer between CCE and FE. The detailed schematic diagram of charge transfer in DC-TENG can also be found in many works (**Supplementary Fig. 8b**)^{26, 28}.

The corresponding output charge and current is shown in **Fig. 2b.**” in the first paragraph of “**SEREF for regulating breakdown domains of DC-TENG**”.

We have added “

28. Chen, C. Y. et al. Direct current fabric triboelectric nanogenerator for biomotion energy harvesting. *ACS Nano* **14**, 4585-4594 (2020).

” In the **Reference** section of the manuscript.

The revised part in the present supplementary information is as follows:

Figure R10 (Revised Supplementary Figure 8). The detailed mechanism for DC-TENG. (a) The simulated result of the potential difference between CCE and FE. Obviously, the potential of FE is higher than that of CCE, thus negative charges transfer from CCE to FE. **(b)** The detailed charge transfer mechanism for DC-TENG.

Remark-2

2. When the FE move from left to right there will be negative triboelectric charges on the TL (triboelectric layer) due to triboelectric effect which can create a negative potential on TL, hence a potential difference between FE and CCE. In figure 2b, author has mentioned that there is no charge transfer between FE and CCE when sliding from right to left. Therefore, authors need to elaborate on how the charges transfer between FE and CCE for a whole unit cycle (right to left and left to right)?

Response

We appreciate your professional review and generous comments. We have made a clear description in the above answer. In short, when DC-TENG moves left, negative charge transfer from CCE to FE because of the potential difference between them (caused by electrostatic breakdown between CCE and TL); when DC-TENG moves right, there is no charge transfer between FE and CCE because of no potential difference between them (no electrostatic breakdown between CCE and TL). The detailed working mechanism can also be found in many previous works, and we also provide a detailed description of the charge transfer mechanism of DC-TENG in our manuscript and supplemented it with detailed schematic diagrams in the supporting information (**Supplementary Fig. 8**).

Remark-3

3. In Figure 3, authors have mentioned that Fig. 3b represents the V_{oc} . If the V_{oc} was measured between FE and CCE, this means that there is (theoretically) no charge transfer from FE to CCE. However, the graph shows that the V_{oc} increases with the increase of this gap between FE and CCE. Can the authors explain the reasons behind this?

Response

We appreciate your professional review and reasonable suggestion. We fully agree with your comment that “If the V_{oc} was measured between FE and CCE, this means that there is (theoretically) no charge transfer from FE to CCE.”. We are very sorry for not clarifying the **theoretical** V_{OC} and **actual** V_{OC} in the previous manuscript.

For the theoretical V_{OC}

Ideally, the open-circuit voltage (V_{OC}) of TENG can be calculated by the equation 5.

$$V_{OC, ideal} = Q_{SC}/C_T \quad (5)$$

where Q_{SC} is the short-circuit charges, and C_T is the inherent capacitor of TENG (**Supplementary Note 2**).

For the achievable V_{OC} in experiment

In actual conditions, the theoretical V_{OC} generally cannot be obtained, which is mainly due to the unique working mechanism of TENG. On one hand, comparing with the large internal resistance and high voltage characteristics of TENG, the measurement instruments often fail to meet the requirements for the open-circuit condition. On the other hand, the unavoidable parasitic capacitance, the charge loss caused by electrostatic breakdown, and even the structural parameters will influence the achievable V_{OC} in experiment, leading to the measured voltage significantly smaller than the theoretical V_{OC} , which have been demonstrated in many previous works^{8,9}. Therefore, the achievable V_{OC} should be defined as the terminal voltage in the open-circuit state. Obviously, there is no charge transfer in external circuit in this state.

For the achievable V_{OC} of DC-TENG

Given that the special structure and working mechanism of DC-TENG, the achievable V_{OC} is more complex for different structures. In this work, the circuit for testing output voltage of TENG is shown in **Figure 2g**. As shown in **Figure R11**, for the DC-TENG with/without insulator, the output voltage fluctuates below a certain value due to the occurrence of 3rd BD. This means that during this period (after output voltage reach this certain value) the charge transferred from DC-TENG to the C_{test} is basically 0, which is very close to the open-circuit state that no charge transfer in external circuit, so we define the certain value as the V_{OC} of DC-TENG. Obviously, the V_{OC} of DC-TENG is equal to the breakdown threshold of 3rd BD, and the observation experiment results (**Figure R11c** and **Figure R11d**) indicate that the spark discharge will cross the air gap between CCE and FE. Therefore, the larger the gap between CCE and FE or the wider the width of the insulator is, the higher the breakdown threshold of the 3rd BD is, thus leading to a higher V_{OC} .

In addition, to eliminate the possibility of the influence of the testing circuit on the experiment results, besides

conducting the leakage analysis experiment of the testing circuit mentioned in the present manuscript (**Figure 2g** and **h**), we also supplemented the experiment to analysis the influence of different testing capacitors on the experiment results (**Figure R12**). It is clearly that the achievable V_{OC} is independent of the test capacitor, indicating that this voltage is the achievable maximum voltage for DC-TENG in experiment. We also added this part in the supplementary information to explain the definition of actual V_{OC} and the difference between actual V_{OC} and ideal V_{OC} , and revised the part of **Supplementary Note 5**.

Figure R11 (Added Supplementary Figure 13.). The breakdown threshold of 3rd BD. (a) The output voltage of DC-TENG without insulator. (b) The output voltage of DC-TENG with insulator. The output voltage of DC-TENG will no longer over the certain value due to the occurrence of 3rd BD. (c) and (d) are the photos of spark discharge of 3rd BD (Scale bar: 5 mm).

Figure R12 (Added Supplementary Figure 12.). The output voltage of DC-TENG when the C_{test} is different. The experimental results indicate that C_{test} has no effect on the 3rd BD of DC-TENG.

Revisions

The revised part in the present manuscript is as follows:

We have made a revision of “From **Fig. 2h**, the charges stored in the C_{test} should be equal to the charges released from the C_{test} , indicating that the test result is not affected by leakage current of the test circuit, which is important for accurate test of output voltage (**Supplementary Fig. 12 and Supplementary Note 5**.” in the last paragraph of “Performance enhancement of different mode DC-TENGs by improving Coulombic efficiency”.

We have made a revision of “Therefore, limited by the 3rd BD, the maximum output voltage of DC-TENG with the insulator is a fixed value (**Fig. 2k, Supplementary Fig. 12-13 and Supplementary Note 5**)” in the last paragraph of “Performance enhancement of different mode DC-TENGs by improving Coulombic efficiency”.

We have made a revision of “To clearly understand the relationship of output energy, Q_{SC} , and V_{OC} , the Q - V curve of TENG with voltage as the horizontal axis and charge as the vertical axis is plotted (**Fig. 3g, Supplementary Fig. 22 and Supplementary Note 5**,” in the second paragraph of “Introducing Coulombic efficiency as a figure-of-merit for accurately evaluating the performance of TENG”.

The revised part in the present Supplementary information is as follows:

we have merged the original “**Supplementary Note 7 The open-circuit voltage of TENG**” and “**Supplementary Note 4 Test method for output voltage of DC-TENG**” to current “**Supplementary Note 5 Definition and testing method for the open-circuit voltage of TENG**”.

Supplementary Note 5 Definition and testing method for the open-circuit voltage of TENG

Ideally, the open-circuit voltage (V_{OC}) of TENG can be calculated by the formula 2.

$$V_{\text{OC, ideal}} = Q_{\text{SC}}/C_{\text{T}} \quad (2)$$

where C_{T} is the inherent capacitor of TENG (**Supplementary Note 2**).

In actual conditions, the theoretical V_{OC} generally cannot be obtained, which is mainly due to the unique working mechanism of TENG. On one hand, comparing with the large internal resistance and high voltage characteristics of TENG, the measurement instruments often fail to meet the requirements for the open-circuit condition. On the other hand, the unavoidable parasitic capacitance, the charge loss caused by electrostatic breakdown, and even the structural parameters will influence the achievable V_{OC} in experiment, leading to the measured voltage significantly smaller than the theoretical V_{OC} , which have been demonstrated in many previous works^{5, 8}. Therefore, the achievable V_{OC} should be defined as the terminal voltage in the open-circuit state. Obviously, there is no charge transfer in external circuit in this state.

The test circuit in **Fig. 2g** is used to accurately test the output voltage of DC-TENG, where the DC-TENG is in series with the test capacitor (C_{test}), protective resistors (R_{p1} and R_{p2}) and the electrometer to form a closed circuit. The electrometer is used to measure the quantity of flowed charges in the loop. Here, the output voltage of DC-TENG ($V_{\text{DC-TENG}}$) can be regarded as:

$$V_{\text{DC-TENG}} = V_{\text{C}} + V_{\text{R}} \quad (3)$$

Where V_C is the voltage of capacitor, V_R is the voltage of R_{p1} .

Given that the output current is very small ($\sim 1 \mu\text{A}$), the V_R is much smaller than V_C . V_R can be neglected, so the following equation can be obtained.

$$V_{\text{DC-TENG}} \approx V_C = \frac{Q}{C_{\text{test}}} \quad (4)$$

where Q is the charge stored in capacitor. Therefore, we can obtain the output voltage of DC-TENG. It is noted that if there is leakage current in the test circuit, the tested charge will be larger than the actual stored charge in the capacitor, and the voltage calculated by equation (3) will be higher than the actual voltage, resulting in inaccurate voltage measurement. To avoid such inaccurate tests, we also measured the released charge from the C_{test} with the S_1 off and S_2 on (**Fig. 2g<ii>**), and the discharge curve can be regressed to zero, which indicates that the tested charge is equal to the actual stored charge in C_{test} . **Finally, in order to eliminate the possibility of the influence of the testing circuit on the experiment results, we also analysis the influence of different testing capacitors on the experiment results (Supplementary Fig. 12).** In other words, this test method is suitable for measuring the output voltage of DC-TENG.

Remark-4

4. The authors mentioned that the average power of the “DC-TENG with insulator” increased by 5.7 times, 7.9 times, and 10.7 times when the insulator width was increased, whereas the voltage increased by 2.2 times, 4.2 times, and 5.9 times (while Q_{sc} was kept stable). Can authors explain here so as to why the power output improvement is not proportional to the voltage increment?

Response

We appreciate your careful review and professional comments.

In general, the output energy of TENG is often considered to be proportional to the square of the charge density, or to the product of the Q_{sc} and the V_{OC} (Supplementary Note 8)^{10, 11}. However, the premise for this conclusion is that regardless of the output voltage, and the surface charge density of TENG must remain constant, which means that electrostatic breakdown does not occur. Actually, electrostatic breakdown is an inevitable experimental phenomenon. Therefore, we speculate that there is still a critical but always neglected factor, which greatly affects the output performance of DC-TENG.

In present work, we introduced Coulombic efficiency as a new figure-of-merit to correctly quantify the output performance of TENGs. We considered that the improvement of Coulombic efficiency is the reason for that the multiple of power increase is much higher than the multiple of voltage increase (Q_{sc} keeps stable) (Detailed discussions can be found in the section titled ‘Introducing Coulombic efficiency as a figure-of-merit for accurately evaluating the performance of TENG’ in the manuscript). The maximum output energy of TENG can be calculated

by Q_{SC} , V_{OC} and Coulombic efficiency as:

if the $\eta(V_{OC}) \leq 50\%$, the maximum output energy can be calculated as:

$$E_{\max} = 50\% \times Q_{SC} \times V_{\eta=50\%} \quad (6)$$

If the $\eta(V_{OC}) \geq 50\%$, the maximum output energy can be calculated as:

$$E_{\max} = \eta_{V_{OC}} \times Q_{SC} \times V_{OC} \quad (7)$$

$V_{\eta=50\%}$ refers to the voltage corresponding to a Coulomb efficiency of 50%.

We calculated the maximum energy output of TENG, and compared them with the maximum output energy tested in the experiment. The comparison results are shown in the **Figure R13**. After introducing Coulomb efficiency, the calculated data are closer to the experimental data.

We have added the reason for the multiple of power increase is much higher than the multiple of voltage increase (Q_{SC} keeps stable) in our revised manuscript.

Revisions

The revised part in the present manuscript is as follows:

We have made a revision of “The results of DC-TENG with different structure in **Fig. 3j** demonstrate that $\eta(V)$ will be much higher than that of DC-TENG without insulator under the same voltage (**Supplementary Fig. 25** and **Supplementary Table 1**). In addition, we calculated the maximum energy output of TENG by Coulomb efficiency, and compared them with the maximum output energy tested in the experiment. The comparison results are shown in the **Supplementary Fig. 26**. After introducing Coulomb efficiency, the calculated results are closer to the experimental results, which also demonstrate our strategy improving the performance of DC-TENG from three aspects: the output charge, the output voltage and Coulombic efficiency.” in the last paragraph of “Introducing Coulombic efficiency as a figure-of-merit for accurately evaluating the performance of TENG”.

The revised part in the present Supplementary information is as follows:

Figure R13 (Added Supplementary Figure 26). The maximum output energy comparison of DC-TENG. (a) The output charge and output voltage of DC-TENG. (b) The output energy of DC-TENG.

Remark-5

5. Can authors explain the motion profiles they used for their experiments, so that the readers are aware of the test conditions? Please include the information about the rate of movement (amplitude, frequency/velocity) as well as the contact force if available.

Response

We appreciate your professional review and reasonable suggestion. We are very sorry for not providing complete testing parameters of our experiments in the previous manuscript. We have added the testing parameters of each experiment in the supporting information, including frequency, amplitude, acceleration, and pressure.

Revisions

The revised part in the present Supplementary information is as follows:

Revised Supplementary Table 8. The detailed motion parameters of sliding-mode TENG

Parameters Device	Acceleration	Deceleration	Maximum rate	Frequency	Distance	Pressure
DC-TENG (Figure 2)	0.2 m s ⁻²	0.2 m s ⁻²	0.1 m s ⁻¹	0.5 Hz	50 mm	10 N
DC-TENG (Figure 2)	0.8 m s ⁻²	0.8 m s ⁻²	0.2 m s ⁻¹	1 Hz	50 mm	10 N
CDC-TENG (Figure 4)	0.8 m s ⁻²	0.8 m s ⁻²	0.2 m s ⁻¹	1 Hz	50 mm	10 N
DEDC-TENG (Figure 4)	0.2 m s ⁻²	0.2 m s ⁻²	0.1 m s ⁻¹	0.5 Hz	50 mm	10 N
AC-TENG (Figure 5)	0.4 m s ⁻²	0.4 m s ⁻²	0.1 m s ⁻¹	1 Hz	25 mm	10 N

Response to Reviewer #3

General remark

The manuscript presents a technological development in the field of triboelectric nanogenerator, which is a relevant topic for the scientific community.

The work is focused on the main characteristics for a long-term operation, which is a proper strategy since it is relevant for practical purposes. Deep understanding of the physical mechanism that limit the development of new technologies play a relevant role in basic science.

Response

We highly appreciate the reviewer for carefully reviewing our work, and thank your positive and generous comments on our research work, which is a great encouragement for us.

Remark-1

As a main backward of this technology is claimed that the discharge due to electrostatic breakdown inducing a loss of power density. It is in this aspect that I will focus my attention on the review of the results.

Response

We appreciate your professional review. As the reviewer pointed out, electrostatic breakdown is indeed an important factor limiting performance optimization of triboelectric nanogenerators (TENGs). Therefore, great efforts were devoted to studying the electrostatic breakdown phenomenon in TENGs. We believe a systematic review of the research background would greatly assist the reviewer in understanding the development direction of this field and clarifying the advancements made in our manuscript.

Research background of electrostatic breakdown in TENGs

Early established standards for evaluating TENG performance emphasize that output performance is directly proportional to the square of charge density for TENGs with identical structural parameters¹¹. Consequently, beyond optimizing structural design to achieve superior structural figure-of-merits^{5, 12-16}, a significant focus for TENG performance enhancement depends on increasing surface charge density. However, electrostatic breakdown often poses a limit on the achievable surface charge density on triboelectric layer, as evidenced by vacuum experiments conducted in 2017 (where TENGs exhibited higher surface charge density under vacuum conditions)¹⁷. Therefore, various methods were proposed to restrict the electrostatic breakdown including environmental control¹⁷⁻²², ultrathin dielectric layer²³⁻²⁶, etc., and the charge density of TENG can be improved a lot.

However, the measured average power density of TENG did not increase in proportion to the square of charge density. Because, previous works naturally considered that the surface charge density keeps stable, i.e., the surface charge density at load conditions is the same as the value in short-circuit condition, and ignored that the electrostatic breakdown also can alter the shape of the $V-Q$ curve (The $V-Q$ curve is used for the purpose of describing energy

output of each motion cycle of TENG.). Here, we discovered the surface charge and power loss resulting from electrostatic breakdown under load conditions, and proved that the slope of the V - Q curve remains constant while the curve itself collapses inward (The slope of the V - Q curve is the reciprocal of the inherent capacitance of the TENG.). In response to this observation, we have made the following advancements in our manuscript.

Advancements made in our manuscript

1. A new mechanism for restricting electrostatic breakdown in TENGs

Conventional strategies restrict electrostatic breakdown by increasing the critical breakdown electric field of breakdown domain (the process 2 in Figure 1a and f), the new mechanism proposed in this paper modulates the electric field strength of breakdown domain below critical breakdown electric field by the spontaneously established reverse electric field (**the process 3 in Figure 1a and f**). This is completely different from the previous works and possesses merits of simple structure, self-regulation behavior and higher performance.

2. A simple and universal method for building the spontaneously established reverse electric field

The specific method explored in this paper is carried out by only pasting an insulator at the electrode edge, which is used to prevent charge leakage due to side-discharge flowing into the electrode, and to spontaneously accumulate static charges. Besides, benefited from the self-regulation of the spontaneously established reverse electric field (SEREF) (the self-regulation of SEREF refers as that the electric field strength of SEREF increases with output voltage or surface charge density), this method not only improves performance of TENG from Q_{SC} and V_{OC} , but is also the first to improve performance from Coulombic efficiency.

3. An unexpected and record-high output energy density

With the demonstration of DC-TENG devices, our strategy not only improves the short-circuit charge (Q_{SC}) and open-circuit voltage (V_{OC}), but also solves the issue of low charge utilization efficiency for the first time, and then a substantial enhancement of average power density of $2.3 \text{ W m}^{-2} \text{ Hz}^{-1}$ (increased by 54 times) is achieved owing to the improvement of Coulombic efficiency from 28.2% to 94.8%. For conventional AC-TENG, a record-breaking average power density of $6.15 \text{ W m}^{-2} \text{ Hz}^{-1}$ is also realized.

4. A new figure-of-merit for accurately evaluating the performance of TENG.

A new figure-of-merit, Coulombic efficiency, is proposed and demonstrated for correctly quantifying the output performance of TENGs, overcoming the issue of surface charge density decline dynamically caused by electrostatic breakdown. It provides a clear direction for guiding the design of high-performance TENGs and should be a standardized parameter for quantifying the performance of TENGs.

5. Huge potential for technology applications.

The performance of TENG can be further optimized by the synergetic enhancement of improved triboelectric charge

density as many previous works reported and improved Coulombic efficiency as our strategy in the future, setting the foundation for the further applications and industrialization of TENG technology.

Remark-2

The analysis is not adequate since it is not correct the claim in the introduction where the authors are assuming that this effect is unexpected and indicating that it is limiting factor for practical power management.

Response

We appreciate your professional comments. We are very sorry for not clarifying this description in the previous manuscript. To eliminate any confusion, we have removed the word of “unexpected” in the revised manuscript, and we also provide a detailed explanation in the revised supplementary information.

Revisions

The revised part in the present manuscript is as follows:

We have made a revision of “The core issue is that the charge loss from electrostatic breakdown (especially side-discharge) at large load voltage is neglected, making the high charge density at short-circuit condition meaningless whether for practical power management or energy storage (**Supplementary Fig. 1-2 and Supplementary Note 1**)” in the second paragraph of “Introduction”.

The revised part in the present Supplementary information is as follows:

Supplementary Note 1 The charge loss in TENG resulted from power management

The intrinsic output characteristics of TENGs are high output voltage (several kilovolts) and low output current (μA), which are not matched with the input requirement of high input current (mA) and only several volts of normal electronic devices (TENG can only achieve optimal output power when the load voltage is high.). Therefore, the power management circuit (PMC) is indispensable to bridge the gap between the TENG and electronic devices. **Supplementary Fig. 1a** shows a commonly used power management circuit in TENG field. The more energy C_{in} ($E_{in} = 0.5 * C_{in} * V_A^2$) stores, the more energy it can be used to drive electronic devices. Therefore, node A has a high potential, which can ensure that TENG generates sufficient output power. Node B has a low potential, which can ensure the normal operation of electronic devices.

The high potential at node A is necessary for efficient power management of TENG. However, with the increase of the potential at node A (It can be controlled by a voltage source in this experiment.), the output charge declines rapidly due to a decrease in the surface charge of triboelectric layer caused by electrostatic breakdown (**Supplementary Fig. 2**). In this paper, we report a novel strategy that the spontaneously established reverse electric field (SEREF) between the electrode and triboelectric layer restricts electrostatic breakdown by decreasing the electric field strength below

critical breakdown electric field, which can be achieved by only pasting an insulator at the electrode edge (Supplementary Fig. 2b).

In addition, the high potential difference between nodes in the power management circuit can also cause electrostatic breakdown, thereby reducing the efficiency of power management circuits. This problem can be solved by encapsulating the power management module, such as circuit encapsulated in a highly insulating epoxy resin to prevent internal breakdown and leakage of electricity¹.

Figure R14 (Added Supplementary Figure 1). The power management of TENG. (a) The represented power management circuit. (b) The output power of DC-TENG with different output voltage with/without power management circuit. The DC-TENG used in this experiment is shown in Fig. 4h. C_{in} is 55 pF, which is obtained by connecting four 220 pF capacitors in series. The threshold voltage of gas discharge tube (GDT) is 1500 V. The reverse breakdown voltage of freewheeling diode is 2000 V. L is 330 μ H. C_{out} is 10 μ F.

Figure R15 (Added Supplementary Figure 2). The output charge of DC-TENG when the potential at node A increases. (a) The test circuit. The potential at node A can be controlled by changing the output voltage of the voltage source. (b) The output charge of DC-TENG. The DC-TENG used in this experiment is shown in Fig. 2j and 2k.

Remark-3

High power management of the BEOL or interconnect technology is well-known in the semiconductor industry, and different techniques and proper design can be implemented to mitigate the early breakdown of the interconnect dielectrics. As an example, different automotive application implements 10kV signal in dielectrics of 10 -20 microns.

Therefore, it is possible to overcome the challenges that are present in this manuscript as a limiting factor.

Response

We appreciate your professional review and reasonable suggestion. As the suggestion of reviewer, we studied some literatures relating to high power management of the BEOL or interconnect technology in the semiconductor industry. After conducting research on these technologies, we have gained a lot of inspiration in suppressing air breakdown in the field of TENG.

Power semiconductor devices are the core devices of power integrated circuits²⁷. When working in a blocking state, a depletion region will be generated near the PN junction and a strong electric field will exist, which may eventually lead to avalanche breakdown²⁸. Junction termination technology (JTT) is often used to mitigate avalanche breakdown caused by excessively high local electric fields in high-power semiconductor devices. It can be categorized into two main classes. The first class involves the introduction of charges near the main junction. These charges create an additional electric field that modulates the electric field distribution, reducing the peak electric field at the main junction and broadening the depletion region width, such as field plates (FP)^{29, 30}, field limiting ring (FLR)³¹ and junction terminal extensions (JTE)^{33, 34}. The second class focuses on removing the junction regions with high curvature and concentrated electric fields. For example, trench termination technology³⁵ can be used to optimize the electric field distribution by etching trenches and filling them with dielectric materials with low permittivity and higher critical electric field strength.

We consider that regardless of the TENG research field (The methods to suppress air breakdown in TENG have been provided in the manuscript and in the answer to **Remark 1**.) or the semiconductor industry, the methods for suppressing breakdown can be categorized as two categories: 1. Reducing the electric field strength in the breakdown region. 2. Increasing the critical electric field strength in the breakdown region.

The strategy proposed in this work for suppressing air breakdown involves introducing charges on the side surface of the insulator and utilizing the electric field established by these charges to modulate the electric field intensity in the air domain (also referred to as the second breakdown domain (2nd BD) of the DC-TENG in the manuscript). It is worth noting that these charges spontaneously accumulate on the side surface of the insulator through air breakdown. This method for modulating the electric field intensity in the breakdown domain has not been reported in the TENG research field.

Furthermore, inspired by the FP technique (When voltage is applied to the field plate, it generates an additional electric field that superimposes with the electric field of the main junction, thereby reducing the electric field intensity near the main junction.), we also introduced an additional metal electrode B at the edge of the existing electrode (referred to as electrode 1 (E1)). Through simulation calculations, we found that the electric field strength at the breakdown region gradually decreases (**Figure R16**) as the voltage applied to electrode 2 (E2) increases. This provides us with another approach to suppressing air breakdown by modulating the electric field strength. However,

as a device for energy harvesting, using an additional voltage source to adjust the voltage applied to electrode B would be redundant for TENG. Therefore, we will try to achieve this objective through the design of the TENG, and we would like to express our gratitude to the reviewer again for the valuable comments.

Figure R16 (Added Supplementary Figure 36). The simulated electric field around the edge of electrode (surface charge density of the triboelectric layer is $-50 \mu\text{C m}^{-2}$). E2 is used to modulate the electric field. The insulator between electrodes is mainly used for electrode isolation. -<iv> As the voltage applied to E2 increases from zero to -500 V , the region of the breakdown domain gradually decreases. The data in (b) is taken from the electric field intensity at $5 \mu\text{m}$ above the triboelectric layer in (a).

Revisions

The revised part in the present manuscript is as follows:

We have made a revision of “As shown in Fig. 1f, conventional strategies (increasing the threshold of electrostatic breakdown from E_{B1} to E_{B2}) optimized TENG’s performance from two aspects: Q_{SC} or V_{OC} . Our strategy focuses on modulating the electric field intensity in the breakdown domain by the SEREF, and comprehensively optimizes TENG’s performance from three aspects: Q_{SC} , V_{OC} and Coulombic efficiency ($\eta(V)$), therefore the output energy can be greatly enhanced. The detailed discussions are shown in the following parts.” in the first paragraph of “Discussion”.

We have made a revision of “It is noteworthy that our strategy for modulating the electric field intensity to suppress electrostatic breakdown in the breakdown domain has not been previously reported in the research field of TENG. Furthermore, we hypothesize that in addition to utilizing surface charge for electric field regulation, adjusting the terminal voltage of the electrode may also serve as a viable method (Supplementary Fig. 36 and Supplementary Note 13). More importantly, the performance of TENGs could be further optimized by the synergetic enhancement

of improved triboelectric charge density as many previous works reported and improved Coulombic efficiency as our strategy in the future.” in the first paragraph of “Discussion”.

The revised part in the present Supplementary information is as follows:

Supplementary Note 13 Modulating the electric field intensity in the breakdown domain

The strategy proposed in this work for suppressing air breakdown involves introducing charges on the side surface of the insulator and utilizing the electric field established by these charges to modulate the electric field intensity in the air domain (also referred to as the second breakdown domain (2nd BD) of the DC-TENG in the manuscript). It is worth noting that these charges spontaneously accumulate on the side surface of the insulator through air breakdown. This method for modulating the electric field intensity in the breakdown domain has not been reported in the TENG research field.

Besides, we speculate that apart from utilizing surface charge to regulate the electric field, adjusting the terminal voltage of electrode is also a viable method. Here, we introduced an additional metal electrode B at the edge of the existing electrode (referred to as electrode 1 (E1)). Through simulation calculations, we found that the electric field strength at the breakdown region gradually decreases (**Supplementary Fig. 36**) as the voltage applied to electrode 2 (E2) increases. This provides us another approach to suppress air breakdown by modulating the electric field strength. However, as a device for energy harvesting, using an additional voltage source to adjust the voltage applied to electrode B would be redundant for TENG.

Remark-4

Moreover, the repetitive loss of charge reported in the manuscript should also be analyzed in terms of the reliability of the proposed devices.

Response

We appreciate your professional review and reasonable suggestion. We have tested the reliability of the device. As shown in **Figure R17**, the output current of the device remains stable after prolonged continuous operation, indicating the good working stability of our devices.

Figure R17 (Added Supplementary Figure 14). The reliability of the DC-TENG.

Then, we analyzed the stability of the charges on the side surface of the insulator. Experimental results indicate that the side surface charges can be maintained at a high value for a long time without external charge supplementation (Figure R18, the testing method is shown in Supplementary Fig. 4. The movement distance is 10 cm.). In addition, the side surface charges can reach saturation in a short time (The movement distance of one cycle is 5 cm, thus side surface charges can reach saturation no more than two motion cycles.). Even if the surface charges decay, it can be quickly recharged. This result provides another strong evidence for the reliability of the side surface charges.

Figure R18 (Added Supplementary Figure 5). The residual charge density on the side surface of insulator without external charge supplementation. After 30 s, 60 s and 90 s, the surface charge retention rate is 87.9%, 76.8% and 76.2%, respectively. The testing method for charge density on the side surface of insulator is shown in Supplementary Fig. 4.

Revisions

The revised part in the present manuscript is as follows:

We have made a revision of “However, massive negative charges will accumulate on the surface of insulator due to electrostatic breakdown during sliding movement (Fig. 1c and Supplementary Fig. 4), and it can be maintained at a high value for a period of time without external charge supplementation (Supplementary Fig. 5).” in the second paragraph of “Concept of the spontaneously established reverse electric field”.

We have made a revision of “The DC-TENG with SEREF exhibits comprehensively enhanced performance and good

reliability (Supplementary Fig. 14).” in the first paragraph of “Performance enhancement of DC-TENG by SEREF”.

Remark-5

A deep understanding of the degradation mechanism during the process of discharge should also be provided.

Response

We appreciate your professional review and reasonable suggestion. We are very sorry for the lacking of the mechanism of discharge in the original manuscript.

The breakdown mechanism of gas is usually based on Townsend avalanche, which is described by the empirical relationship between breakdown voltage (V_b) and the product of gap distance (d) and gas pressure (p). Generally, the breakdown criterion is derived based on two important parameters: the electron impact ionization coefficient α and the secondary electron emission coefficient γ .

$$\gamma(e^{\alpha d} - 1) = 1 \tag{8}$$

where α describes the generation of ions by electron impact and the corresponding equation is as follows.

$$\alpha = Ape^{-Bpd/V_b} \tag{9}$$

where A, B are two constants relating to gas compositions, and p is the gas pressure. Thus, the following equation can be obtained.

$$V_b = \frac{Bpd}{\ln(Apd) + \ln\left[\frac{1}{\ln\left(\frac{1}{\gamma} + 1\right)}\right]} \tag{10}$$

This is the Paschen curve, and the Paschen’s constants for air is: $p, 101.25 \text{ kPa}$; $A, 10.95 \text{ (m}\times\text{Pa)}^{-1}$; $B, 273.8 \text{ V (m}\times\text{Pa)}^{-1}$; $\gamma, 8.136\times 10^{-3}$.

Given that the electric field at the air gap of contact-separation TENG (CS-TENG) is approximately considered as a uniform field, Paschen curve can be used to determine whether air breakdown occurs in CS-TENG, and can also be used to calculate the maximum surface charge density of CS-TENG (Figure R19).

Figure R19 (Added Supplementary Figure 34). The breakdown theory of CS-TENG. (a) The structure diagram of CS-TENG. V_{air} is the air gap voltage of CS-TENG. σ is the surface charge density of triboelectric layer. t and ϵ_r are the thickness and relative dielectric constant of triboelectric layer, respectively. ϵ_0 is $8.85 \times 10^{-12} \text{ F m}^{-1}$. **(b)** The relationship between V_b curve (the orange line) and V_{air} (the blue lines). When σ is $250 \mu\text{C m}^{-2}$, t is $50 \mu\text{m}$, ϵ_r is 2.1, air breakdown does not occur due to two curves are tangent, and $250 \mu\text{C m}^{-2}$ is the maximum surface charge density. If increasing σ , increasing t or decreasing ϵ_r , then the two curves intersect and air breakdown occurs. If decreasing σ , decreasing t or increasing decreasing ϵ_r , then the two curves separate and air breakdown does not occur.

For the sliding mode TENG, the Paschen's law cannot be directly used for calculation, because of the non-ideal conditions. For example, the electric field around the sliding mode TENG is not uniform as assumed in Paschen's law, which depends on the mechanical configuration, triboelectric materials, surface roughness, and other issues. Irrespective of the specific circumstance, there always exists a fixed breakdown threshold for the device that is fabricated (The breakdown could occur at the gap between the electrodes, the gap between the electrodes and the triboelectric layer, etc.).

Besides the demonstration methods in the original manuscript, we also provide another method to demonstrate the mechanism of discharge of sliding mode TENG. The device used in this experiment is the general sliding mode alternating current TENG (AC-TENG) (**Figure R20a**). The two bottom electrodes remain fixed, and the triboelectric layers are PVDF (polyvinylidene difluoride), PI (polyimide), ETFE (ethylene-terafluoroethylene), PTFE (Polytetrafluoroethylene), PVC (polyvinyl chloride) and FEP (fluorinated ethylene propylene), respectively. Firstly, we used the strategy proposed in this work to suppress side-discharge of electrodes (**Figure R20b **) (Here, the insulating material is the same as the material of triboelectric layer to avoid triboelectrification between these two materials as much as possible.), and measured the surface charge density of triboelectric layer (the purple points in **Figure R20c**). Then, we removed the insulating material between the two bottom electrodes (**Figure R20b <ii>**), and measured the surface charge density of triboelectric layer again (the orange points in **Figure R20c**). It is obviously that the surface charge density of triboelectric layer decays to a low fixed value regardless of the triboelectric materials. These results demonstrate that the breakdown threshold is fixed when the device structure is determined again.

Figure R20 (Added Supplementary Figure 35). The output charge of AC-TENG with/without insulator. (a) The structure diagram of AC-TENG. **(b)**  AC-TENG with insulator.  AC-TENG without insulator. **(c)** The surface charge density of AC-TENG.

In addition to the Townsend avalanche, other effects such as field emission of electrons and tunneling electrons may also exist. However, the electric field intensity threshold for triggering the field emission is $\sim 75 \text{ MV m}^{-1}$, which is higher by an order of magnitude for triggering Townsend avalanche. Therefore, we consider that the performance of TENG is ultimately limited by Townsend avalanche in the atmospheric environment.

We also added this part as **Supplementary Note 12** in supplementary information.

Revisions

The revised part in the present manuscript is as follows:

We have made a revision of “Beyond of DC-TENGs based on triboelectrification and electrostatic breakdown, the conventional AC-TENGs based on triboelectrification and electrostatic induction also suffers from the serious side-discharge problem (Fig. 5a, Supplementary Fig. 33-35 and Supplementary Note 12).” in the first paragraph of “Performance enhancement of AC-TENG by improving Coulombic efficiency”.

The revised part in the present Supplementary information is as follows:

Supplementary Note 12 Breakdown theory

The breakdown mechanism of gas is usually based on Townsend avalanche, which is described by the empirical relationship between breakdown voltage (V_b) and the product of gap distance (d) and gas pressure (p). Generally, the breakdown criterion is derived based on two important parameters: the electron impact ionization coefficient α and the secondary electron emission coefficient γ .

$$\gamma(e^{\alpha d} - 1) = 1 \quad (21)$$

where α describes the generation of ions by electron impact and the corresponding equation is as follows.

$$\alpha = Ape^{-Bpd/V_b} \quad (22)$$

where A , B are two constants relating to gas compositions, and p is the gas pressure. Thus, the following equation can be obtained.

$$V_b = \frac{Bpd}{\ln(Apd) + \ln\left[\frac{1}{\ln\left(\frac{1}{\gamma} + 1\right)}\right]} \quad (23)$$

This is the Paschen curve, and the Paschen's constants for air is: p , 101.25 kPa; A , $10.95 \text{ (m}\times\text{Pa)}^{-1}$; B , $273.8 \text{ V (m}\times\text{Pa)}^{-1}$; γ , 8.136×10^{-3} .

Given that the electric field at the air gap of contact-separation TENG (CS-TENG) is approximately considered as a uniform field, Paschen curve can be used to determine whether air breakdown occurs in CS-TENG, and can also be used to calculate the maximum surface charge density of CS-TENG (**Supplementary Fig. 34**).

For the sliding mode TENG, the Paschen's law cannot be directly used for calculation, because of the non-ideal conditions. For example, the electric field around the sliding mode TENG is not uniform as assumed in Paschen's law, which depends on the mechanical configuration, triboelectric materials, surface roughness, and other issues. Irrespective of the specific circumstance, there always exists a fixed breakdown threshold for the device that is fabricated (The breakdown could occur at the gap between the electrodes, the gap between the electrodes and the triboelectric layer, etc.).

Besides the demonstration methods in the original manuscript, we also provide another method to demonstrate the mechanism of discharge of sliding mode TENG. The device used in this experiment is the general sliding mode alternating current TENG (AC-TENG) (**Supplementary Fig. 35a**). The two bottom electrodes remain fixed, and the triboelectric layer are PVDF (polyvinylidene difluoride), PI (polyimide), ETFE (ethylene-terafluoroethylene), PTFE (Polytetrafluoroethylene), PVC (polyvinyl chloride) and FEP (fluorinated ethylene propylene) . Firstly, we used the strategy proposed in this work to suppress side-discharge of electrodes **Supplementary Fig. 35b ** (Here, the insulating material is the same as the material of triboelectric layer to avoid triboelectrification between these two materials as much as possible.), and measured the surface charge density of triboelectric layer (the purple points in **Supplementary Fig. 35c**). Then, we removed the insulating material between the two bottom electrodes (**Supplementary Fig. 35b <ii>**), and measured the surface charge density of triboelectric layer again (the orange points in **Supplementary Fig. 35c**). It is obviously that the surface charge density of triboelectric layer decays to a very low fixed value regardless of the triboelectric materials. These results demonstrate that the breakdown threshold is fixed when the device structure is determined again.

Remark-6

Based on this analysis I suggest considering publication of the manuscript after a mayor review in terms of the physics behind the discharge of the storage charge considering the state of the art of interconnect technology. I recommend

the inclusion of cross section TEM to confirm the dimension of the DUTs.

Response

We appreciate your professional review and reasonable suggestion. Given the dimension of our device, preparing the TEM samples are very difficult, so we provided the cross-section SEM for demonstration (**Figure R21**). In addition, we also analyzed the element distribution at the observation area by using the energy-spectrum scanning function of SEM to provide a clearer perspective. We would like to express our gratitude to the reviewer again for the valuable comments.

Revisions

The revised part in the present manuscript is as follows:

We have made a revision of “To suppress the electrostatic breakdown of 2nd BD, here an insulator is introduced to past at the FE edge (**Fig. 2c**), which can utilize side-discharge around FE to accumulate charges on insulator’s surface and then generates a reverse electric field to suppress electrostatic breakdown in 2nd BD. **The cross section of DC-TENG is shown in Supplementary Fig. 10.**” in the third paragraph of “SEREF for regulating breakdown domains of DC-TENG”.

The revised part in the present Supplementary information is as follows:

Figure R21 (Added Supplementary Figure 10). The cross section of the DC-TENG.  The structure diagram of DC-TENG (a) without insulator, (b) with insulator. The purple shaded area is the observation area.  SEM image of the observed area (scale bar: 200 μm).  The distribution of elements at the observation area of DC-TENG was analyzed by using the energy-spectrum scanning function of SEM. The yellow dots represent copper element, green dots represent carbon element. Due to the devices are handmade and the cross section is relatively rough, it is difficult to distinguish the positional relationship between electrodes and insulator solely based on SEM images. Therefore, we have supplemented the element distribution at the observation area. Due to the substrate of the device is acrylic (Polymeric Methyl Methacrylate, PMMA) and the insulation layer is polyimide (PI), their main element is carbon and does not contain copper. The main element of the electrode is copper and does not contain carbon. Therefore, the electrodes and insulator can be distinguished by analyzing the distribution of carbon and copper elements at the observation area.

Reference

1. Bi, M. Z. et al. Optimization of structural parameters for rotary freestanding-electret generators and wind energy harvesting. *Nano Energy* **75**, 104968 (2020).
2. Li, Q. Y. et al. Overall performance improvement of direct-current triboelectric nanogenerators by charge leakage and ternary dielectric evaluation. *Energy Environ. Sci.* **16**, 3514-3525 (2023).
3. He, L. X., et al. A high-output silk-based triboelectric nanogenerator with durability and humidity resistance. *Nano Energy* **108**, 108244 (2023).
4. He, L. X. et al. A dual-mode triboelectric nanogenerator for wind energy harvesting and self-powered wind speed monitoring. *ACS Nano* **16**, 6244-6254 (2022).
5. Niu, S. M. et al. Theoretical study of contact-mode triboelectric nanogenerators as an effective power source. *Energy Environ. Sci.* **6**, 3576-3583 (2013).
6. Zhao, Z. H. et al. Selection rules of triboelectric materials for direct-current triboelectric nanogenerator. *Nat. Commun.* **12**, 4686 (2021).
7. He, W. C. et al. Boosting output performance of sliding mode triboelectric nanogenerator by charge space-accumulation effect. *Nat. Commun.* **11**, 4277 (2020).
8. Gao, Y. K. et al. Achieving high-efficiency triboelectric nanogenerators by suppressing the electrostatic breakdown effect. *Energy Environ. Sci.* **16**, 2304-2315 (2023).
9. Liu, J. Q. et al. Achieving ultra-high voltage ($\approx 10\text{kV}$) triboelectric nanogenerators. *Adv. Energy Mater.* **13**, 2300410 (2023).
10. Zi, Y. L. et al. Effective energy storage from a triboelectric nanogenerator. *Nat. Commun.* **7**, 10987 (2016)..
11. Zi, Y. L. et al. Standards and figure-of-merits for quantifying the performance of triboelectric nanogenerators. *Nat. Commun.* **6**, 8376 (2015).
12. Niu, S. M. et al. A theoretical study of grating structured triboelectric nanogenerators. *Energy Environ. Sci.* **7**, 2339-2349 (2014).
13. Niu, S. M. et al. Theory of sliding-mode triboelectric nanogenerators. *Adv. Mater.* **25**, 6184-6193 (2013).
14. Niu, S. M. et al. Theoretical investigation and structural optimization of single-electrode triboelectric nanogenerators. *Adv. Funct. Mater.* **24**, 3332-3340 (2014).
15. Wang, S. H., Niu, S. M., Yang, J., Lin, L. & Wang, Z. L. Quantitative measurements of vibration amplitude using a contact-mode freestanding triboelectric nanogenerator. *ACS Nano* **8**, 12004-12013 (2014).
16. Wang, S. H., Xie, Y. N., Niu, S. M., Lin, L. & Wang, Z. L. Freestanding triboelectric-layer-based nanogenerators for harvesting energy from a moving object or human motion in contact and non-contact modes. *Adv. Mater.* **26**, 2818-2824 (2014).
17. Wang, J. et al. Achieving ultrahigh triboelectric charge density for efficient energy harvesting. *Nat. Commun.* **8**, 88 (2017).
18. Zhou, L. L. et al. Simultaneously enhancing power density and durability of sliding-mode triboelectric nanogenerator via interface liquid lubrication. *Adv. Energy Mater.* **10**, 2002920 (2020).
19. Yi, Z. Y. et al. Enhancing output performance of direct-current triboelectric nanogenerator under controlled atmosphere. *Nano Energy* **84**, 105864 (2021).
20. Zi, Y. L., Wu, C. S., Ding, W. B. & Wang, Z. L. Maximized effective energy output of contact-separation-triggered triboelectric nanogenerators as limited by air breakdown. *Adv. Funct. Mater.* **27**, 1700049 (2017).
21. Xu, G. Q. et al, Environmental lifecycle assessment of CO₂-filled triboelectric nanogenerators to help achieve carbon neutrality. *Energy Environ. Sci.* **16**, 2112-2119 (2023).
22. Liu, D. et al. Hugely enhanced output power of direct-current triboelectric nanogenerators by using electrostatic breakdown effect. *Adv. Mater. Technol.* **5**, 2000289 (2020).

23. Zhang, C. L. et al. Surface charge density of triboelectric nanogenerators: Theoretical boundary and optimization methodology. *Adv. Mater. Technol.* **18**, 100496 (2020).
24. Li, Y. H. et al. Improved output performance of triboelectric nanogenerator by fast accumulation process of surface charges. *Adv. Energy Mater.* **11**, 2100050 (2021).
25. Liu, W. L. et al. Integrated charge excitation triboelectric nanogenerator. *Nat. Commun.* **10**, 1426 (2019).
26. Liu, Y. K. et al. Quantifying contact status and the air-breakdown model of charge-excitation triboelectric nanogenerators to maximize charge density. *Nat. Commun.* **11**, 1599 (2020).
27. Huang, A. Q. Power semiconductor devices for smart grid and renewable energy systems. *Proc. IEEE* **105**, 2019-2047 (2017).
28. Qiao, M. et al. A review of high-voltage integrated power device for AC/DC switching application. *Microelectron. Eng.* **232**, 111416 (2020).
29. Hirao, T., Onose, H., Yasui, K. & Mori, M. Edge termination with enhanced field-limiting rings insensitive to surface charge for high-voltage SiC power devices. *IEEE Trans. Electron Devices* **67**, 2850-2853 (2020).
30. Dora, Y. et al. High breakdown voltage achieved on AlGaN/GaN HEMTs with integrated slant field plates. *IEEE Electron Device Lett.* **27**, 713-715 (2006).
31. Deng, X. et al. Multizone gradient-modulated guard ring technique for ultrahigh voltage 4H-SiC devices with increased tolerances to implantation dose and surface charges. *IEEE J. Emerging Sel. Top. Power Electron.* **7**, 1505-1512 (2019).
32. Onose, H., Yamada, R., Mori, M., Kobayashi, Y. & Onuki, J. Communication—realization of DC bias reliability by 7-zone JTE termination technology. *ECS J. Solid State Sci. Technol.* **5**, Q271 (2016).
33. Huang, C. F. et al. Counter-doped JTE, an edge termination for HV SiC devices with increased tolerance to the surface charge. *IEEE Trans. Electron Devices* **62**, 354-358 (2015).
34. Zhang, L. et al. Low-loss SOI-LIGBT with triple deep-oxide trenches. *IEEE Trans. Electron Devices* **64**, 3756-3761 (2017).

REVIEWERS' COMMENTS

Reviewer #1 (Remarks to the Author):

The authors have addressed my questions and they are acceptable.

Reviewer #2 (Remarks to the Author):

The authors have addressed the Reviewer's concerns adequately in the revised manuscript and added significant modifications. Therefore, this manuscript can be recommended for publication.

Reviewer #3 (Remarks to the Author):

The authors have analyzed all the comments from my previous review. I am satisfied with their answers, since in all cases all the aspects of the comments were covered.

The focus of my present review is the loss of power density due to discharge by electrostatic breakdown.

A systematic review was included to clarify the root cause of the work of the present manuscript. In this way, the loss of power is not presented as a limiting factor but as a characteristic of the TENG. A deeper analysis of the charge loss mechanism was also included to explain the practical power management application.

Regarding the power management of BEOL interconnections, the quality of the manuscript has increased significantly. A deep discussion including references was included in the manuscript. Moreover, I consider that the inclusion of this topic enhances the quality of the manuscript since the development of innovative devices is strongly correlated with the limitations of interconnect technology.

The authors clarified the main aspects of the variability under continuous operation, showing good perspectives for the reliability of such devices.

Finally a new section was included analyzing the structure of the devices under test. The cross-section SEM figures (with the energy-spectrum scanning) provide information about the dimensions and material involved.

Based on these corrections, I think that the manuscript should be considered for publication.

Point-by-point responses to the reviewers' comments

We sincerely thank the reviewer's positive evaluation of our work. Their prior expert professional comments and suggestions have significantly strengthened and smoothed out our work. The following responses are prepared to address all of the reviewers' comments in a point-by-point fashion. (**Comments in Black, responses in Blue**)

REVIEWER COMMENTS

Reviewer #1 (Remarks to the Author):

The authors have addressed my questions and they are acceptable.

Response:

We highly appreciate the reviewer for carefully reviewing our work, and thank your valuable comments on our research work and recommendation for publications.

Reviewer #2 (Remarks to the Author):

The authors have addressed the Reviewer's concerns adequately in the revised manuscript and added significant modifications. Therefore, this manuscript can be recommended for publication.

Response:

We highly appreciate the reviewer for carefully reviewing our work, and thank your positive comments on our research work and recommendation for publications.

Reviewer #3 (Remarks to the Author):

The authors have analyzed all the comments from my previous review. I am satisfied with their answers, since in all cases all the aspects of the comments were covered.

The focus of my present review is the loss of power density due to discharge by electrostatic breakdown. A systematic review was included to clarify the root cause of the work of the present manuscript. In this way, the loss of power is not presented as a limiting factor but as a characteristic of the TENG. A deeper analysis of the charge loss mechanism was also included to explain the practical power management application.

Regarding the power management of BEOL interconnections, the quality of the manuscript has increased significantly. A deep discussion including references was included in the manuscript. Moreover, I consider that the inclusion of this topic enhances the quality of the manuscript since the development of innovative devices is strongly correlated with the limitations of interconnect technology.

The authors clarified the main aspects of the variability under continuous operation, showing good perspectives for the reliability of such devices.

Finally a new section was included analyzing the structure of the devices under test. The cross-section SEM figures (with the energy-spectrum scanning) provide information about the dimensions and material involved.

Based on these corrections, I think that the manuscript should be considered for publication.

Response:

We highly appreciate the reviewer for carefully reviewing our work, and thank your contribution to our research work and recommendation for publications.